# The phosphatase DUSP22 inhibits UBR2-mediated K63-ubiquitination and activation of Lck downstream of TCR signalling

Ying-Chun Shih[1], Hsueh-Fen Chen[1], Chia-Ying Wu[1], Yi-Ru Ciou[1], Chia-Wen Wang[1], Huai-Chia Chuang[1] ✉ & Tse-Hua Tan [1] ✉

DUSP22 is a dual-specificity phosphatase that inhibits T cell activation by inactivating the kinase Lck. Here we show that the E3 ubiquitin ligase UBR2 is a positive upstream regulator of Lck during T-cell activation. DUSP22 dephosphorylates UBR2 at specific Serine residues, leading to ubiquitin-mediated UBR2 degradation. UBR2 is also modified by the SCF E3 ubiquitin ligase complex via Lys48-linked ubiquitination at multiple Lysine residues. Single-cell RNA sequencing analysis and UBR2 loss of function experiments showed that UBR2 is a positive regulator of proinflammatory cytokine expression. Mechanistically, UBR2 induces Lys63-linked ubiquitination of Lck at Lys99 and Lys276 residues, followed by Lck Tyr394 phosphorylation and activation as part of TCR signalling. Inflammatory phenotypes induced by TCR-triggered Lck activation or knocking out DUSP22, are attenuated by genomic deletion of UBR2. UBR2-Lck interaction and Lck Lys63-linked ubiquitination are induced in the peripheral blood T cells of human SLE patients, which demonstrate the relevance of the UBR2-mediated regulation of inflammation to human pathology. In summary, we show here an important regulatory mechanism of T cell activation, which finetunes the balance between T cell response and aggravated inflammation.

Lymphocyte cell-specific protein-tyrosine kinase (Lck), a Src-family tyrosine kinase, is phosphorylated and activated in the propagation of T-cell receptor (TCR) signaling. Lck kinase activity is dynamically regulated through conformational changes by phosphorylation and dephosphorylation of its tyrosine residues[1,2]. Phosphorylation of Lck Tyr505 by C-terminal Src kinase (CSK) results in a closed and inactive conformation of Lck in resting T cells[1,2]. Dephosphorylation of Lck Tyr505 by CD45 results in an open conformation of Lck, leading to trans-autophosphorylation of Tyr394 and subsequent activation of Lck[1,2]. Lys48-linked ubiquitination of Lck by the E3 ubiquitin ligase Cbl leads to proteasomal degradation of Lck[3]. It was unclear whether Lck kinase activity is regulated by ubiquitination (such as Lys63-linked ubiquitination).

JNK pathway-associated phosphatase (JKAP, also named DUSP22) belongs to the dual-specificity phosphatase (DUSP)

family[4,5]. DUSPs dephosphorylate threonine/serine and/or tyrosine residues of their substrates[6,7]. Dysregulation of DUSP22 in T cells contributes to the development of several human diseases[7], including peripheral T-cell lymphoma[8], inflammatory bowel disease (IBD)[9], systemic lupus erythematosus (SLE)[10], and ankylosing spondylitis (AS)[11]. DUSP22 dephosphorylates and inactivates focal adhesion kinase, resulting in the suppression of cancer cell motility and liver cancer progression[12,13]. DUSP22 dephosphorylates the tyrosine kinase Lck at Tyr394 in the turn-off stage of TCR signaling, leading to the suppression of T cell-mediated immune responses[14]. Conversely, DUSP22-deficient T cells display induction of the Th1 cytokines IFN-γ and the Th17 cytokine IL-17[10,11,14,15]. Thus, DUSP22 is an important inhibitor of T-cell activation by dephosphorylating and inhibiting Lck in the attenuation phase of TCR signaling.

[1]Immunology Research Center, National Health Research Institutes, Zhunan, Taiwan. ✉e-mail: cinth@nhri.edu.tw; ttan@nhri.edu.tw

In the ubiquitin-proteasome pathway, E3 ubiquitin ligases ubiquitinate the substrate with Lys48-linked ubiquitin chains, targeting the substrate for proteasomal degradation. SKP1−CUL1−F-box proteins (SCF) E3 ubiquitin ligase complexes catalyze the ubiquitination of proteins targeted for 26 S proteasomal degradation[16]. In SCF complexes, the F-box proteins are the substrate-recognition subunits. SKP1 is an adaptor protein essential for the binding of F-box proteins. CUL1 is the major structural scaffold of the SCF complexes. RBX1 (RING-box protein 1) or RBX2 (RING-box protein 2), containing a RING domain, interacts with an E2 ubiquitin-conjugating enzyme and transfers ubiquitin from E2 to a lysine residue on target proteins[15].

Ubiquitin Protein Ligase E3 Component N-Recognin 2 (UBR2), a RING domain E3 ubiquitin ligase, is a component of the N-end rule pathway of targeted proteolysis[17]. UBR2 restricts mobilization of LINE-1 (L1) retrotransposons by inducing the protein ubiquitination and proteasomal degradation of L1-ORF1p[18]. UBR2 also induces the protein degradation of the cleaved N-terminal NLRP1B fragment, leading to inflammasome activation[19]. In addition, UBR2 is associated with cancer progression[18,20,21], cancer cachexia autoimmune pancreatitis[22], amyotrophic lateral sclerosis[23], and T-cell chronic lymphocytic leukemia[24]. The predominant expression of UBR2 in immune cells[25] suggests a role of UBR2 in regulating immune cell functions; however, the roles and functions of UBR2 in lymphocytes have not been investigated.

Here, we report that DUSP22 dephosphorylates UBR2 and blocks UBR2-mediated Lck K63-ubiquitination and activation, leading to the reduction of T-cell activation and cytokine production. Our findings on the regulation of Lck activation by UBR2 may provide therapeutic approaches to treat autoimmune diseases.

## Results

### DUSP22 induces UBR2 proteasomal degradation

To search for regulators that control DUSP22 protein levels, anti-Myc-DUSP22 immunocomplexes from transfected HEK293T cells were precipitated and then subjected to proteomics. The E3 ubiquitin-protein ligase UBR2 was identified as a DUSP22-interacting protein (Fig. 1a). We first examined whether UBR2 induces DUSP22 protein degradation. Myc-DUSP22 and Flag-UBR2 plasmids were co-transfected into HEK293T cells, and cell extracts were subjected to immunoblotting analysis. UBR2 overexpression did not affect DUSP22 protein levels (Fig. 1b). Surprisingly, DUSP22 overexpression resulted in a dose-dependent decrease of UBR2 protein levels (Fig. 1c). To test whether DUSP22 phosphatase activity is required for the induction of UBR2 degradation, the DUSP22 phosphatase-dead mutant Myc-DUSP22 (C88S)[14] and Flag-UBR2 plasmids were co-transfected into HEK293T cells. The result showed that overexpression of DUSP22 mutant (C88S) did not inhibit, but increased, UBR2 protein levels in a dose-dependent manner (Fig. 1d), suggesting that DUSP22 phosphatase activity is required for the induction of UBR2 degradation. To examine whether UBR2 protein levels are increased by DUSP22 knockout, we generated DUSP22 knockout mice by TALEN-mediated gene targeting (Fig. 1e). DUSP22 knockout was confirmed by PCR and immunoblotting analyses (Fig. 1f). The deletion of 4 bp (GATC) in DUSP22 exon 1 resulted in a 24-amino acid frame-shift/truncated mutant (Fig. 1e). The mutant proteins were not detected by an anti-DUSP22 antibody that recognized the C-terminus of murine DUSP22 proteins (Fig. 1g). Although UBR2 is predominantly expressed in immune cells[25], UBR2 proteins were detected in T cells and multiple non-lymphoid tissues of wild-type mice (Fig. 1g). The protein levels of UBR2 in multiple tissues of DUSP22 knockout mice were increased compared to those of wild-type mice (Fig. 1g). The purified T cells of DUSP22 knockout mice also showed increased UBR2 protein levels compared to those of wild-type mice (Fig. 1g). These results support that DUSP22 induces UBR2 degradation. In addition to immune cells, UBR2 protein may also play a role in other tissues or cell types.

To study whether DUSP22 induces UBR2 degradation by a caspase-, lysosome-, or proteasome-mediated pathway, cells were treated with the caspase inhibitor Z-VAD-FMK, the lysosomal enzyme inhibitor chloroquine, or the proteasome inhibitors MG132 and carfilzomib. The data showed that DUSP22-induced UBR2 proteasomal degradation was reversed by treatment of MG132 (Fig. 1h) or carfilzomib (Supplementary Fig. 1a); however, the degradation was not affected by Z-VAD-FMK or chloroquine treatment (Fig. 1i). The data suggest that DUSP22 induces UBR2 proteasomal degradation. We next confirmed the interaction between DUSP22 and UBR2 in cells using reciprocal coimmunoprecipitation assays. Flag-UBR2 plasmid was co-transfected with either Myc-DUSP22 or Myc-DUSP22 (C88S) plasmid into HEK293T cells. The transfected cell lysates were subjected to immunoprecipitation with either anti-Flag antibody (Fig. 1j) or anti-Myc antibody (Fig. 1k), followed by immunoblotting. These assays confirmed the interaction between DUSP22 and UBR2 proteins, as well as between DUSP22 (C88S) mutant and UBR2 proteins (Fig. 1j, k). As expected, the levels of coimmunoprecipitated UBR2 proteins by DUSP22 substrate-trapping (C88S) mutant[26] were higher than that of wild-type DUSP22 (Fig. 1j, k). This result also ruled out the possibility that the DUSP22 (C88S) mutant[26] reversed UBR2 degradation due to its inability in binding to UBR2. To further demonstrate the interaction between DUSP22 and UBR2 proteins in vivo, we performed in situ proximity ligation assay (PLA), which detects two molecules in close proximity (< 40 nm) in cells with paired antibody-conjugated probes[27]. The data showed strong PLA signals of DUSP22-UBR2 interaction in MG132-treated Jurkat T cell but not in other negative control T cells (Fig. 1l). We further examined whether DUSP22 (C88S) mutant competes with wild-type DUSP22 for UBR2 binding. The PLA data showed that the interaction between DUSP22 and UBR2 proteins in HEK293T cells was decreased by DUSP22 (C88S) overexpression (Supplementary Fig. 1b). To confirm the direct interaction of DUSP22 with UBR2, in vitro binding assays were performed using purified proteins. Purified Flag-tagged UBR2 proteins and recombinant GST-tagged DUSP22 proteins were subjected to in vitro binding assays. The results showed that GST-tagged DUSP22 proteins strongly interacted with Flag-tagged UBR2 proteins (Fig. 1m). Collectively, these results showed that DUSP22 directly interacted with UBR2.

### Lys94, Lys779, and Lys1599 residues of UBR2 are responsible for DUSP22-induced UBR2 degradation

Next, to study whether DUSP22 induces Lys48-linked ubiquitination of UBR2, we immunoprecipitated Flag-tagged UBR2 proteins from HEK293T cells co-transfected with Flag-UBR2 and either Myc-DUSP22 or vector control plasmids. The data showed that DUSP22 overexpression enhanced UBR2 Lys48-linked ubiquitination (Fig. 2a). Heterotypic Lys11-linked ubiquitination also serves as a targeting signal for proteasome[28], thus we also tested whether DUSP22 enhances UBR2 Lys11-linked ubiquitination. Flag-tagged UBR2 proteins were immunoprecipitated from HEK293T cells co-transfected with Lys11-only ubiquitin (K11-Ub) mutant, Flag-UBR2, and either Myc-DUSP22 or vector control plasmids. The data showed that DUSP22 overexpression did not enhance UBR2 Lys11-linked ubiquitination (Supplementary Fig. 2a). To confirm that these DUSP22-induced ubiquitinated protein bands were UBR2 itself but not UBR2-interacting proteins, we performed two rounds of immunoprecipitations. The anti-Flag-UBR2 immunoprecipitates were denatured using 1% SDS and boiling to dissociate UBR2-interacting proteins, the proteins were renatured through serial dilution, and the renatured Flag-UBR2 proteins were re-immunoprecipitated. Ubiquitinated UBR2 bands were still detected in the second-round anti-Flag-UBR2 immunoprecipitates (Fig. 2b), indicating that UBR2 itself is ubiquitinated. To study whether DUSP22 is responsible for inducing UBR2 Lys48-linked ubiquitination, we depleted DUSP22 using DUSP22 shRNAs and then assessed the levels of UBR2 ubiquitination. DUSP22 shRNA knockdown decreased Lys48-

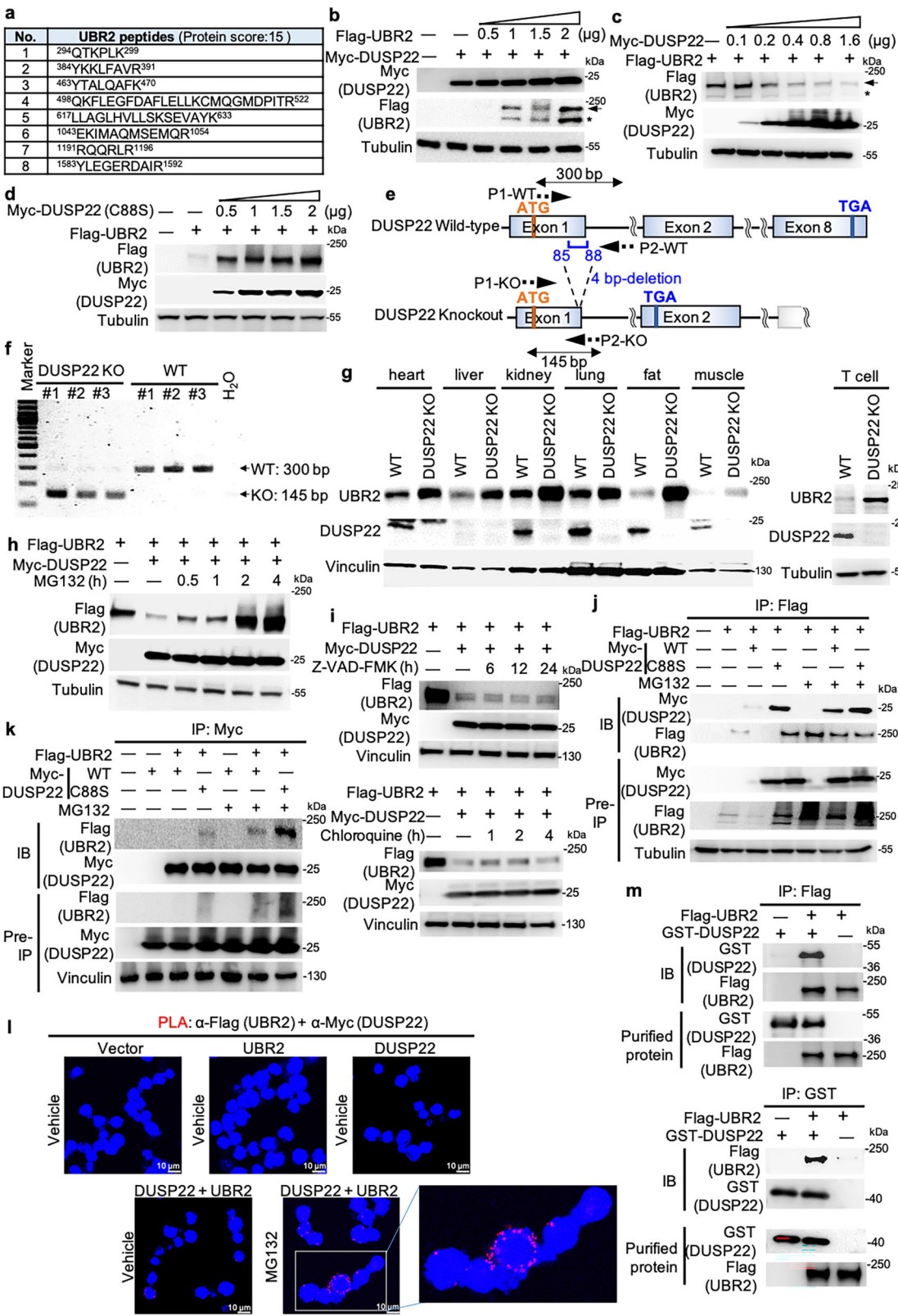

linked ubiquitination of UBR2 (Fig. 2c). Furthermore, DUSP22 (C88S) overexpression reduced Lys48-linked ubiquitination of UBR2 (Supplementary Fig. 2b), suggesting that DUSP22 phosphatase activity is required for the induction of UBR2 Lys48-linked ubiquitination. To identify the DUSP22-induced Lys48-linked ubiquitination sites responsible for UBR2 degradation, we immunoprecipitated Flag-tagged UBR2 proteins from HEK293T cells co-transfected with Flag-

UBR2, Lys48-only ubiquitin (K48-Ub) mutant, and Myc-DUSP22 plasmids. The immunoprecipitated Flag-tagged UBR2 proteins were then subjected to mass spectrometry analysis. The result showed that UBR2 was ubiquitinated at Lys94 only when co-transfected with Lys48-only ubiquitin (K48-Ub) plus DUSP22 under MG132 treatment (Fig. 2d and Supplementary Table 1). The Lys255 residue was ubiquitinated when co-transfected with Lys48-only ubiquitin (K48-Ub) in the absence of

**Fig. 1 | DUSP22 induces UBR2 degradation. a** The identified peptide sequences of endogenous UBR2 protein by mass spectrometry-based analyses using Myc-tagged immunocomplexes from the lysate of Myc-DUSP22-transfected HEK293T cells. **b-d** Immunoblotting of Flag-tagged UBR2, Myc-tagged DUSP22, and tubulin proteins in HEK293T cells co-transfected with different amounts of Flag-UBR2 plus either Myc-DUSP22 (**b** and **c**) or Myc-DUSP22 (C88S) (**d**) plasmids. Tubulin immunoblotting was performed by reprobing the Flag (UBR2) immunoblot membrane (**b** and **d**). Arrow, intact UBR2 protein; asterisk, degraded UBR2 protein. **e** Schematic diagram of the murine DUSP22 wild-type (WT) alleles and the targeted DUSP22 mutant alleles. P1 and P2, the primers for PCR. **f** Characterization of DUSP22-knockout mice. PCR analyses of wild-type and DUSP22 mutant alleles in the genomic DNA from mouse tails. The PCR product of the upper band (300 bp) denotes the wild-type allele, and the lower band (145 bp) denotes the DUSP22 mutant allele. **g** Immunoblotting analysis of UBR2 protein levels in multiple tissues of wild-type and DUSP22 knockout mice. **h, i** DUSP22-induced UBR2 proteasomal degradation. Flag-UBR2 and Myc-DUSP22 plasmids were co-transfected into

HEK293T cells. The transfected cells were treated with MG132 (25 μM), Z-VAD-FMK (50 μM), or chloroquine (50 μM) for the indicated time points and then subjected to immunoblotting analysis. Anti-tubulin immunoblotting was performed by reprobing the anti-Myc (DUSP22) immunoblot membrane (**h**). Vinculin immunoblotting was performed by reprobing the anti-Flag (UBR2) immunoblot membrane (**i**). **j, k** UBR2 interacted with DUSP22 or DUSP22 (C88S) mutant proteins. Coimmunoprecipitation and immunoblotting analyses of UBR2, DUSP22, and DUSP22 (C88S) mutant proteins in HEK293T cells co-transfected with Flag-UBR2 plasmid plus either Myc-DUSP22 or Myc-DUSP22 (C88S) mutant plasmid. **l** Proximity ligation assays (PLA) showed in vivo UBR2-DUSP22 interaction in Jurkat T cells. Red fluorescence represents interactions (< 40 nm) of Flag-UBR2 and Myc-DUSP22 proteins. Images were captured by confocal microscope (Leica TCS SP5 II). Original magnification, 400X. Cell nucleus was stained with DAPI. Scale bar, 10 μm. **m** In vitro binding assays of purified Flag-tagged UBR2 and recombinant GST-tagged DUSP22 proteins.

DUSP22, but undetectable in the presence of DUSP22, suggesting that Lys255 residue was not responsible for DUSP22-induced UBR2 ubiquitination (Supplementary Fig. 3a and Supplementary Table 1). We also analyzed the mass spectrometry data from the experiment using HEK293T cells transfected with Flag-UBR2 and Myc-DUSP22 plasmids without Lys48-only ubiquitin (K48-Ub) mutant plasmid. The result showed that UBR2 was ubiquitinated at 9 lysine residues, including Lys158, Lys165, Lys248, Lys470, Lys488, Lys568, Lys789, Lys1496, and Lys1689 (Supplementary Fig. 3b; Supplementary Table 2). Besides mass spectrometry data, Lys731, Lys798, Lys958, and Lys1142 residues were predicted as UBR2 ubiquitination sites using web-based softwares[29] (UbiSite, http://csb.cse.yzu.edu.tw/UbiSite/). PhosphoSitePlus (https://www.phosphosite.org/homeAction) website shows additional 14 UBR2 ubiquitination sites; however, the functional consequences of these ubiquitination sites have not been studied. To study whether ubiquitination of any of the 24 mapped and 4 web-predicted residues induces DUSP22-mediated UBR2 degradation, these lysine residues were individually mutated to arginine residues. Individual UBR2 (K94R), UBR2 (K779R), and UBR2 (K1599R) mutations significantly blocked DUSP22-mediated UBR2 degradation (Fig. 2e). In the absence of DUSP22 overexpression, the expression of each UBR2 mutant alone was comparable to that of wild-type UBR2 (Supplementary Fig. 3c). Furthermore, UBR2 triple mutations (K94/779/1599R) blocked DUSP22-induced UBR2 degradation in Jurkat T cells (Fig. 2f). The Lys48-linked ubiquitination of UBR2 (K94/779/1599R) mutant was abolished compared to that of wild-type UBR2 in HEK293T cells (Fig. 2g). The data suggest that Lys48-linked ubiquitination at Lys94, Lys779, and Lys1599 residues of UBR2 are responsible for DUSP22-mediated UBR2 degradation.

## CUL1-βTrCP is identified as an E3 ubiquitin ligase complex of UBR2

To identify the E3 ubiquitin ligase of UBR2, we searched for UBR2-interacting proteins using HEK293T cells transfected with Flag-UBR2. UBR2-interacting proteins were coimmunoprecipitated with anti-Flag antibody and followed by mass spectrometry analysis. We found that CUL1, a component of SKP1-CUL1-F-box protein (SCF) E3 ubiquitin ligase complex[30], coimmunoprecipitated with UBR2 (Fig. 3a). To study whether CUL1 induces UBR2 degradation, Flag-UBR2 was co-transfected with Myc-DUSP22 and Flag-CUL1 plasmids into Jurkat T cells. The data showed that CUL1 attenuated UBR2 protein levels in a dose-dependent manner (Fig. 3b); however, without DUSP22 over-expression, CUL1 was unable to induce UBR2 degradation (Supplementary Fig. 4a). Next, we verified the interaction between UBR2 and CUL1 proteins using coimmunoprecipitation assays. CUL1 was coimmunoprecipitated with Flag-tagged UBR2 proteins with anti-Flag antibody (Supplementary Fig. 4b). To study the role of CUL1 in DUSP22-induced UBR2 degradation, we examined whether CUL1

knockdown blocks DUSP22-induced UBR2 degradation. The data showed that CUL1 shRNA knockdown reversed DUSP22-induced UBR2 degradation (Fig. 3c). Next, we depleted CUL1 and then assessed the levels of UBR2 ubiquitination. Knockdown of CUL1 reduced UBR2 Lys48-linked ubiquitination (Fig. 3d). These results suggest that CUL1 is responsible for DUSP22-induced UBR2 ubiquitination and degradation.

F-box proteins are the substrate-binding component of SCF E3 ubiquitin ligase complex. The F-box proteins directly bind to substrates and determine the specificity of the E3 ubiquitin ligase complex. βTrCP, SKP2, and FBXW7 are the best-characterized F-box proteins within the CUL1 SCF ubiquitin ligase complexes[15,31]. To study which of these three F-box proteins recognizes UBR2, we performed PLA using HEK293T cells co-transfected with Flag-UBR2 plasmid and an individual plasmid encoding Myc-F-box proteins. The data showed a direct interaction of Flag-tagged UBR2 proteins with Myc-tagged βTrCP proteins but not with Myc-tagged CUL1, SKP2, or FBXW7 proteins (Supplementary Fig. 4c). The interaction of UBR2 with βTrCP was further enhanced by MG132 treatment (Supplementary Fig. 4c). We did not detect any βTrCP tryptic peptides using LC-MS/MS analysis; this may be due to either the limitation of mass spectrometry sensitivity or the inadequate βTrCP peptide lengths generated by trypsin digestion. Next, we examined whether βTrCP induces UBR2 degradation. The result showed that UBR2 protein levels were decreased by βTrCP overexpression in the presence of suboptimal amounts of DUSP22 (Fig. 3e). Although we could not rule out the contribution of additional F-box proteins in controlling UBR2 stability, our data suggest that βTrCP is a key F-box protein for DUSP22-induced UBR2 degradation. To further confirm the interaction of UBR2 with βTrCP, Myc-tagged βTrCP was immunoprecipitated with anti-Myc antibody and then immunoblotted with anti-Flag antibody. The result showed that βTrCP interacted with UBR2 (Fig. 3f). To study whether DUSP22 induces the interaction of UBR2 with βTrCP, PLA assay was performed on HEK293T cells co-transfected with Flag-UBR2, Myc-βTrCP, and either GFP-DUSP22 or GFP-DUSP22 (C88S) mutant plasmids. Interestingly, the data showed that the interaction of Flag-tagged UBR2 proteins with Myc-tagged βTrCP proteins was enhanced by DUSP22 but reduced by DUSP22 (C88S) mutant proteins (Fig. 3g). The subcellular locations where UBR2 interacted with βTrCP coincided with the locations of GFP-tagged DUSP22 proteins (Fig. 3g). We further studied the role of βTrCP in DUSP22-induced UBR2 degradation using shRNA knockdown approach. βTrCP shRNA knockdown blocked DUSP22-induced UBR2 degradation (Supplementary Fig. 4d). To study whether the degradation of UBR2 is driven by βTrCP-CUL1-containing E3 complex during the late stage of T-cell activation, we performed βTrCP or CUL1 shRNA knockdown in Jurkat T cells. The data showed that UBR2 protein levels were decreased at 15 min in the TCR signaling turn-off stage (Fig. 3h). In contrast, UBR2 degradation was blocked by shRNA knockdown of

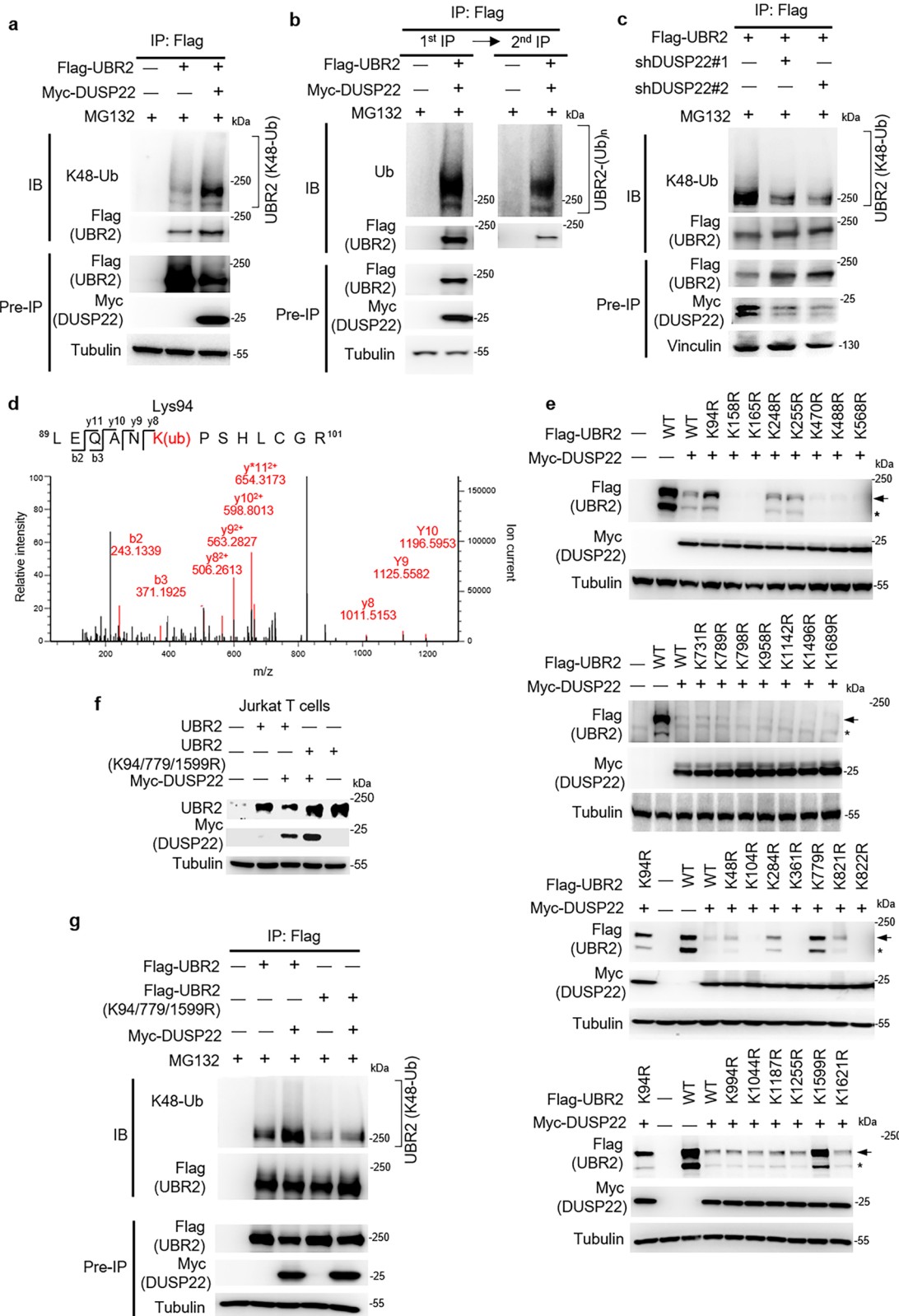

βTrCP or CUL1 in the TCR turn-off stage (Fig. 3h). These results suggested that UBR2 stability is reduced by the βTrCP-CUL1-containing E3 complex during the late stage of T-cell activation. To examine whether UBR2 is ubiquitinated by SKP1-CUL1-βTrCP E3 ubiquitin ligase complex, we performed in vitro ubiquitination assays. To avoid UBR2 ubiquitination by itself, we mutated the UBR2 RING domain by replacing Cys1210 and Cys1213 residues to alanine residues (C1210/1213A).

E3 ligase-inactive Flag-tagged UBR2 (C1210/1213A) proteins were immunoprecipitated with anti-Flag antibody; the protein complexes containing SKP1, RBX1, Myc-tagged βTrCP, and Myc-tagged CUL1 proteins were immunoprecipitated with anti-Myc antibody. After purification by peptide elution, purified Flag-tagged UBR2 (C1210/1213A) proteins were coincubated with the SKP1-CUL1-βTrCP-RBX1 complex plus E1 (UBA1), E2 (UBE2D3 or CDC34), and His-ubiquitin at

**Fig. 2 | DUSP22 induces UBR2 Lys48-linked ubiquitination at Lys94, Lys779, and Lys1599 residues. a** DUSP22 induced UBR2 Lys48-linked ubiquitination. UBR2 was immunoprecipitated with anti-Flag antibody and then immunoblotted with anti-ubiquitin (Lys48) antibody or anti-Flag antibody. **b** UBR2 was ubiquitinated. UBR2 was first immunoprecipitated with anti-Flag antibody (1st IP); half of anti-Flag immunoprecipitates were denatured, renatured, followed by a second immunoprecipitation (2nd IP) with anti-Flag antibody. Anti-Flag immunoprecipitates were immunoblotted with anti-ubiquitin antibody or anti-Flag antibody. **c** DUSP22 knockdown reduced UBR2 Lys48-linked ubiquitination. Flag-UBR2 and individual DUSP22 shRNAs were co-transfected into HEK293T cells. The cells were treated with 25 μM MG132. UBR2 was immunoprecipitated with anti-Flag antibody and then immunoblotted with anti-ubiquitin (Lys48) antibody or anti-Flag antibody. Vinculin immunoblotting was performed by reprobing the Flag (UBR2) immunoblot membrane with anti-vinculin antibody. **d** Lys94 residue was identified as a K48-linked ubiquitination site of UBR2. HEK293T cells were transfected with Flag-UBR2 plus Lys48-only ubiquitin mutant (K48-Ub) plasmids and with or without DUSP22 plasmid, followed by treatment with or without 25 μM MG132 for 4 h. Flag-tagged

UBR2 proteins were immunoprecipitated with anti-Flag antibody and then subjected to mass spectrometry analysis. The MS/MS fragmentation spectra displayed UBR2 tryptic peptides containing the ubiquitinated residue Lys94. K(ub) denotes the ubiquitinated lysine residue. **e** K94R, K779R, or K1599R mutation of UBR2 reversed DUSP22-mediated degradation of UBR2. Myc-DUSP22 and Flag-UBR2 mutant plasmids were co-transfected into HEK293T cells. The levels of Flag-tagged UBR2 proteins were examined by immunoblotting analysis. **f** Triple mutations (K94/779/1599R) of UBR2 blocked DUSP22-mediated degradation of UBR2. Myc-DUSP22 plus either Flag-UBR2 or Flag-UBR2 (K94/779/1599R) mutant plasmids were co-transfected into Jurkat T cells. The levels of UBR2 proteins were determined by immunoblotting analysis. **g** Triple mutations (K94/779/1599R) of UBR2 reduced UBR2 Lys48-linked ubiquitination. Myc-DUSP22 plus either Flag-UBR2 or Flag-UBR2 mutant (K94/779/1599R) plasmids were co-transfected into HEK293T cells. Flag-tagged UBR2 wild-type (WT) or (K94/779/1599R) mutant proteins were immunoprecipitated with anti-Flag antibody and then immunoblotted with anti-ubiquitin (Lys48) antibody or anti-Flag antibody. Arrow, intact UBR2 protein; asterisk, degraded UBR2 protein.

37 °C for 1 h. The data showed that the SKP1-CUL1-βTrCP-RBX1 complex induced ubiquitination of UBR2 (C1210/1213A) in vitro (Fig. 3i), indicating that UBR2 is directly ubiquitinated by CUL1-βTrCP E3 ubiquitin ligase.

## DUSP22 induces UBR2 degradation by dephosphorylating its Ser1694 and Tyr1697 residues

UBR2 ubiquitination was induced by DUSP22 overexpression (Fig. 2a); DUSP22 phosphatase activity was required for UBR2 degradation (Fig. 1d). These results suggest that DUSP22 dephosphorylates UBR2, leading to UBR2 ubiquitination. We studied whether DUSP22 directly dephosphorylates UBR2 using in vitro phosphatase assays and purified proteins. The data showed that Myc-tagged DUSP22 wild-type proteins dephosphorylated Flag-tagged UBR2 at serine and tyrosine residues in vitro (Fig. 4a), whereas the DUSP22 phosphatase-dead mutant (C88S) did not dephosphorylate UBR2 (Fig. 4a). The results indicate that DUSP22 directly dephosphorylates UBR2. To identify the UBR2 dephosphorylation site(s) by DUSP22, Flag-tagged UBR2 proteins were immunoprecipitated from Flag-UBR2-overexpressed HEK293T cells with or without Myc-DUSP22, and characterized by mass spectrometry analysis. Three residues (Ser476, Ser1694, and Tyr1697) of UBR2 were identified to be phosphorylated in the Flag-UBR2 alone group (Fig. 4b), and the three identified phosphorylation sites were undetectable in UBR2-DUSP22 coexpressing cells (Fig. 4b). To study whether the identified phosphorylation sites are targeted by DUSP22, the UBR2 mutants were generated in which Ser476, Ser1694, or Tyr1697 residue was individually mutated to aspartic acid residue, mimicking its phosphorylation state. Compared to UBR2 wild-type, the UBR2 phosphomimetic mutant (S1694D) or UBR2 (Y1697D), but not UBR2 (S476D), resisted DUSP22-induced UBR2 degradation (Fig. 4c). UBR2 double mutations (S1694D/Y1697D) also completely abolished DUSP22-induced UBR2 degradation in HEK293T cells (Fig. 4c). Conversely, the protein stability of UBR2 phosphorylation-deficient mutant (S1694A/Y1697F) was decreased in HEK293T cells (Fig. 4d, left) and Jurkat T cells (Fig. 4d, right) compared to that of wild-type UBR2 (Fig. 4d). The MG132-induced protein stability of UBR2 phosphorylation-deficient mutant (S1694A/Y1697F) was significantly reduced compared to that of wild-type UBR2 (Fig. 4d, left). The data suggest that DUSP22-induced UBR2 dephosphorylation at Ser1694 and Tyr1697 residues, but not Ser476 residue, is responsible for DUSP22-induced UBR2 proteasomal degradation. To study whether the phosphomimetic mutation of UBR2 reduces the DUSP22-mediated UBR2–βTrCP interaction, the PLA assay was performed on HEK293T cells co-transfected with Myc-βTrCP, GFP-DUSP22, as well as individual Flag-UBR2, Flag-UBR2 (S1694D/Y1697D), and Flag-UBR2 (S1694A/Y1697F) plasmids. The data showed that the DUSP22-induced interaction of βTrCP proteins with UBR2 proteins was reduced by

phosphomimetic mutation (S1694D/Y1697D) of UBR2, but further enhanced by phosphorylation-deficient mutation (S1694A/Y1697F) of UBR2 (Fig. 4e). Next, we studied whether the phosphomimetic double mutations of UBR2 (S1694D/Y1697D) block DUSP22-mediated UBR2 ubiquitination. Flag-UBR2 or Flag-UBR2 phosphomimetic mutant (S1694D/Y1697D) plus vector or Myc-DUSP22 plasmids were co-transfected into His-Ub (K48)-transfected or MG132-treated HEK293T cells. Consistently, Lys48-linked ubiquitination of UBR2 phosphomimetic mutant (S1694D/Y1697D) was significantly decreased compared to that of wild-type UBR2 (Fig. 4f). These results suggest that the phosphomimetic mutant (S1694D/Y1697D) of UBR2, resembling its phosphorylated state, would not be efficiently ubiquitinated and targeted for degradation by the SKP1-CUL1-βTrCP complex. In contrast, the phospho-deficient mutant (S1694A/Y1697F) of UBR2 resembles unphosphorylated UBR2 proteins after dephosphorylation by DUSP22, ensuing its degradation by the SKP1-CUL1-βTrCP complex. Collectively, these results indicate that DUSP22 dephosphorylates UBR2 at its Ser1694 and Tyr1697 residues, leading to SCF-CUL1-βTrCP–mediated ubiquitination and subsequent proteasomal degradation of UBR2.

## Single-cell RNA sequencing analysis of UBR2-deficient T cells reveals that UBR2 induces proinflammatory cytokine production

To further investigate the in vivo role of UBR2, we generated UBR2 knockout mice in C57BL/6 J background by CRISPR/Cas9 gene editing (Fig. 5a). UBR2 knockout was confirmed by PCR and immunoblotting analyses (Fig. 5b, c). Homozygous UBR2 knockout male mice may be infertile[32]; we found that homozygous UBR2 knockout male mice were able to produce offspring albeit with very low frequency. The UBR2 knockout mice were produced by intercrossing male UBR2−/− homozygotes with female UBR+/− heterozygotes. In the past 4 years, we obtained only 40 UBR2 homozygous knockout mice, including 38 male and 2 female mice, from 468 offspring mice.

UBR2 is predominantly expressed in immune cells[25], suggesting an important role of UBR2 in immune cell functions. To determine whether T-cell development is affected by UBR2 ablation, thymocytes were isolated from wild-type and UBR2 knockout mice. The data showed that T-cell development and B-cell development were normal in UBR2 knockout mice (Supplementary Fig. 5a–d). The development of spleen-derived myeloid cells was also unaffected by UBR2 deficiency (Supplementary Fig. 5e). To study the T-cell function of UBR2, peripheral blood T cells of wild-type and UBR2 knockout mice were subjected to single-cell RNA sequencing (scRNA-seq) analysis. After quality control filtering, a total of 3061 single-cell transcriptomes from two pairs of mice were identified (Fig. 5d), and 7 major clusters were classified by UMAP analysis (Fig. 5e). Based on individual surface marker

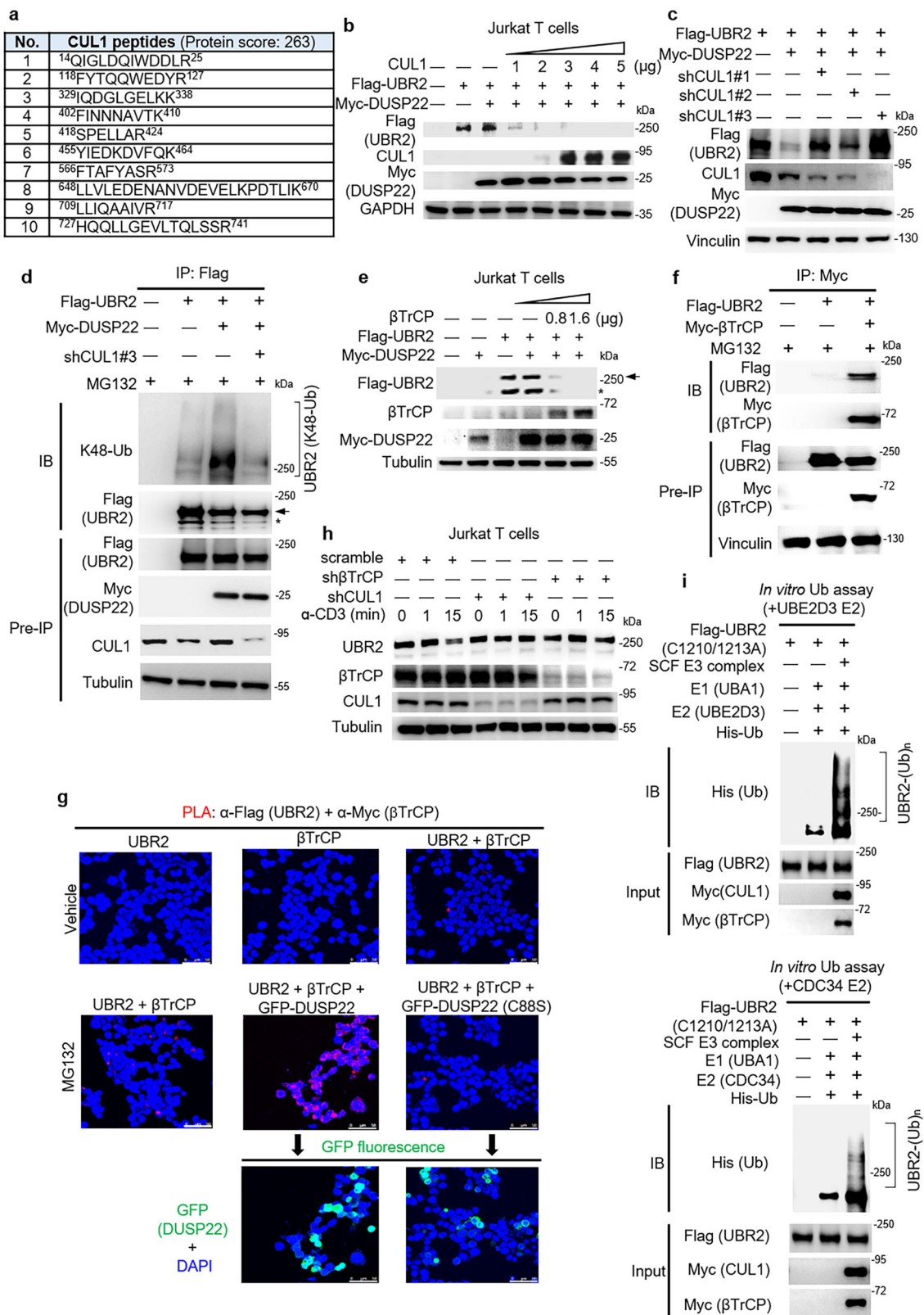

genes of various T-cell subsets (Fig. 5e), (i) Clusters 1, 2, 3, and 6 contained naïve cells (CD44⁻CD62L⁺), (ii) Clusters 5 and 7 contained effector memory T cells ($T_{EM}$, CD44⁺CD62L⁻), and (iii) Cluster 4 contained central memory T cells ($T_{CM}$). Moreover, volcano plot showed that mRNA levels of the T-cell activation markers *Cd28* and *Cd69* in UBR2 knockout T cells were decreased compared to those of wild-type T cells (Fig. 5f). Next, we studied whether UBR2 levels are correlated

with the levels of inflammatory cytokines using scRNA-seq data. We analyzed these correlation using scRNA-seq data derived only from wild-type mice, because UBR2 knockout T cells still expressed non-functional UBR2 mRNAs containing 68-bp internal deletion. The increase of UBR2 mRNA transcripts was correlated with the increase of gene expression of *Ifn-γ* and *Tnf-α* but not *Il-2*, *Il-4*, and *Il-6* (Fig. 5g). To determine whether UBR2 contributes to the induction of

**Fig. 3 | CUL1-βTrCP E3 ligase complex induces Lys48-linked ubiquitination and degradation of UBR2. a** Identification of CUL1 as an UBR2-interacting protein by mass spectrometry-based proteomics. Identified CUL1 peptides were shown. **b** Immunoblotting of CUL1, Flag-tagged UBR2, Myc-tagged (DUSP22), and GAPDH proteins in Jurkat T cells co-transfected with Myc-CUL1, Flag-UBR2, and Myc-DUSP22 plasmids. **c** Immunoblotting of the endogenous CUL1, Flag-tagged UBR2, Myc-tagged DUSP22, and vinculin proteins in HEK293T cells co-transfected with Flag-UBR2, Myc-DUSP22, and CUL1 shRNA plasmids. **d** CUL1 knockdown inhibited DUSP22-induced UBR2 ubiquitination. Immunoprecipitation and immunoblotting analysis of Lys48-linked ubiquitination of UBR2, Flag-tagged UBR2, Myc-tagged DUSP22, and endogenous CUL1 proteins were performed using the lysates of HEK293T cells co-transfected with Flag-UBR2, Myc-DUSP22, and CUL1 shRNA #3 plasmids. The transfected cells were treated with 25 μM MG132 for 4 h. Arrow, UBR2 protein; asterisk, degraded UBR2 protein. **e** βTrCP overexpression plus suboptimal DUSP22 (0.8 μg) induced UBR2 degradation. Immunoblotting of Flag-tagged UBR2, Myc-tagged DUSP22, and tubulin proteins in Jurkat T cells co-transfected with Flag-UBR2 and Myc-DUSP22 plus different amounts (0.8 μg, 1.6 μg) of Flag-βTrCP

plasmids. **f** UBR2 interacted with βTrCP. Immunoprecipitation and immunoblotting of Flag-tagged UBR2 with Myc-tagged-βTrCP proteins were performed using the lysates of HEK293T transfected with Flag-UBR2 and Myc-βTrCP plasmids. Anti-vinculin immunoblotting was performed by reprobing the anti-Flag (UBR2) immunoblot membrane. **g** Confocal microscopy analyses of PLA for the interaction between Flag-tagged UBR2 and Myc-tagged βTrCP proteins in HEK293T cells co-transfected with Flag-UBR2, Myc-βTrCP, and either GFP-DUSP22 or GFP-DUSP22 (C88S) plasmids. Red fluorescence represents the interactions of UBR2 with βTrCP proteins. Images were captured with 400X original magnification. Cell nuclei were stained with DAPI. Scale bar, 50 μm. **h** Immunoblotting of endogenous UBR2, βTrCP, or CUL1 proteins in Jurkat T cells transfected with scramble shRNAs, βTrCP shRNAs #1, or CUL1 shRNAs #3. T cells were stimulated with anti-CD3 antibody for indicated time periods. **i** CUL1-βTrCP E3 ligase complex induced UBR2 ubiquitination in vitro. Recombinant His-ubiquitin, E1 (UBA1), E2 (top panel, UBE2D3; bottom panel, CDC34), ATP, and SKP1-CUL1-βTrCP-RBX1 complex were co-incubated in the ubiquitination buffer with Flag-tagged UBR2 E3 ligase-inactive mutant (C1210/1213A) proteins.

proinflammatory cytokines, we measured the proinflammatory cytokines secreted from anti-CD3 and anti-CD28 costimulated primary T cells of wild-type and UBR2 knockout mice. TCR-stimulated cytokines production of IFN-γ, TNF-α, and IL-17A in T cells of UBR2 knockout mice were decreased compared to those of wild-type T cells (Fig. 5h), suggesting that UBR2 induces the production of proinflammatory cytokines upon TCR signaling. To study whether T-cell differentiation is affected by UBR2 knockout, in vitro differentiation assays were performed using splenic T cells of UBR2 knockout mice. Th1 and Th17, but not Treg, differentiation was decreased in T cells of UBR2 knockout mice compared to wild-type mice (Supplementary Fig. 5g). Collectively, these results suggest that UBR2 plays an important role in Th1/Th17-mediated inflammation.

## UBR2 induces Lck activation by K63-ubiquitinating Lys99 and Lys276 residues of Lck

DUSP22 deficiency results in enhanced TCR signaling[14]. Thus, the effect of UBR2 on the regulation of TCR signaling was investigated using UBR2-knockout murine primary T cells and UBR2-knockdown Jurkat T cells. The tyrosine phosphorylation levels of TCR proximal signaling molecules, such as CD3ζ, Lck, ZAP-70, and LAT were decreased in anti-CD3-stimulated UBR2-knockout primary T cells and UBR2-knockdown Jurkat T cells, decelerating the induction of PLCγ, which is a major molecule connecting the TCR proximal to TCR distal signaling proteins[33], including ERK (Supplementary Fig. 6a, b). Notably, UBR2 levels were not upregulated by α-CD3 stimulation during the TCR signaling turn-on stage (Supplementary Fig. 6a, b). DUSP22 dephosphorylates Lck at its Tyr394 residue during the turn-off stage of TCR signaling, leading to the termination of T-cell signaling[14]. We further studied whether UBR2 is either an upstream regulator or a downstream target of Lck during T-cell activation. The interaction between the endogenous Lck and UBR2 proteins in murine primary T cells was higher at 0 min and 5 min during α-CD3 stimulation (during TCR signaling turn-on stage) compared to those at 10 min, and the Lck-UBR2 interaction was significantly decreased at 10 min (during TCR signaling turn-off stage) (Fig. 6a). In contrast, the Lck-DUSP22 interaction in murine primary T cells was significantly induced at 10 min (during TCR signaling turn-off stage) (Fig. 6a). These results suggested that the Lck-UBR2 interaction precedes the Lck-DUSP22 interaction during the course of α-CD3 signaling. Immunoprecipitation of lysates from Lck-overexpressing Jurkat T cells showed similar results (Supplementary Fig. 6c). The data suggest that the E3 ligase UBR2 may induce Lck ubiquitination in TCR signaling. Lys63-linked (K63) ubiquitination is involved in the regulation of immune responses[34]. We next studied whether Lys63-linked ubiquitination of Lck is induced in TCR signaling. Lys63-linked ubiquitination of Lck was indeed induced upon anti-CD3

stimulation, while tyrosine (Tyr394) phosphorylation of Lck was concomitantly induced in primary T cells of wild-type mice and Jurkat T cells (Fig. 6b and Supplementary Fig. 6d). In addition, Lys48-linked ubiquitination of Lck was not induced in the TCR signaling turn-on stage upon MG132 treatment (Supplementary Fig. 6e). To examine Lys63-linked ubiquitination and tyrosine phosphorylation of Lck in murine primary T cells upon anti-CD3 stimulation, a combination of PLA probes corresponding to anti-Lck antibody plus either anti-Lys63-ubiquitin antibody or anti-phospho-Lck (Tyr394) antibody were used to detect K63-ubiquitinated or Tyr394-phosphorylated Lck. The data showed that Lys63-linked ubiquitination and Tyr394 phosphorylation of Lck were induced by TCR stimulation in T cells of wild-type mice (Fig. 6c and Supplementary Fig. 6f–i), whereas Lys63-linked ubiquitination and Tyr394 phosphorylation of Lck were obliterated by UBR2 knockout (Fig. 6c and Supplementary Fig. 6f–i). Lck-deficient Jurkat (J.Cam1.6 clone) T cells were used as a control to confirm that the observed PLA signals for TCR-induced K63-ubiquitinated or Tyr394-phosphorylated Lck proteins were specific to Lck in TCR signaling but not due to nonspecific background noise (Supplementary Fig. 6h, i). These results suggest that UBR2-induced Lys63-linked ubiquitination of Lck is required for TCR-induced Lck activation. These results support that the interaction between Lck and UBR2 is increased during TCR signaling turn-on stage, leading to UBR2-induced Lys63-linked ubiquitination and activation of Lck. In contrast, the interaction between Lck and DUSP22 is induced in the TCR signaling turn-off stage, resulting in DUSP22-induced dephosphorylation and inactivation of Lck[14]. It is likely that either the catalytic activity or the substrate (Lck)-binding ability of UBR2 could be induced during the TCR signaling turn-on stage.

To verify whether Lck is directly ubiquitinated by UBR2 in vitro, purified Flag-tagged UBR2 or UBR2 RING domain (C1210/1213A) catalytically inactive mutant proteins were coincubated with Flag-tagged Lck plus E1 (UBE1), E2 (UBE2D3), and His-ubiquitin proteins. The in vitro ubiquitination assays showed that wild-type UBR2, but not UBR2 (C1210/C1213A) mutant, induced ubiquitination of Lck in vitro (Fig. 6d), suggesting that Lck is directly ubiquitinated by UBR2 E3 ubiquitin ligase. To identify the Lys63-linked ubiquitination sites of Lck induced by UBR2 in T cells, the immunoprecipitated Flag-tagged Lck proteins from Jurkat T cells were subjected to mass spectrometry analysis. Lys99 and Lys276 residues were identified as the UBR2-targeted ubiquitination resides of Lck (Fig. 6e). We next modeled human Lck protein structure using the online homology modeling server of SWISS-MODEL. Interestingly, Lys99 is located in the SH3 domain of Lck, while Lys276 is located in the kinase domain of Lck (Supplementary Fig. 7), suggesting that Lys63-linked ubiquitination disrupts the closed conformation of Lck, leading to constitutively active state of Lck.

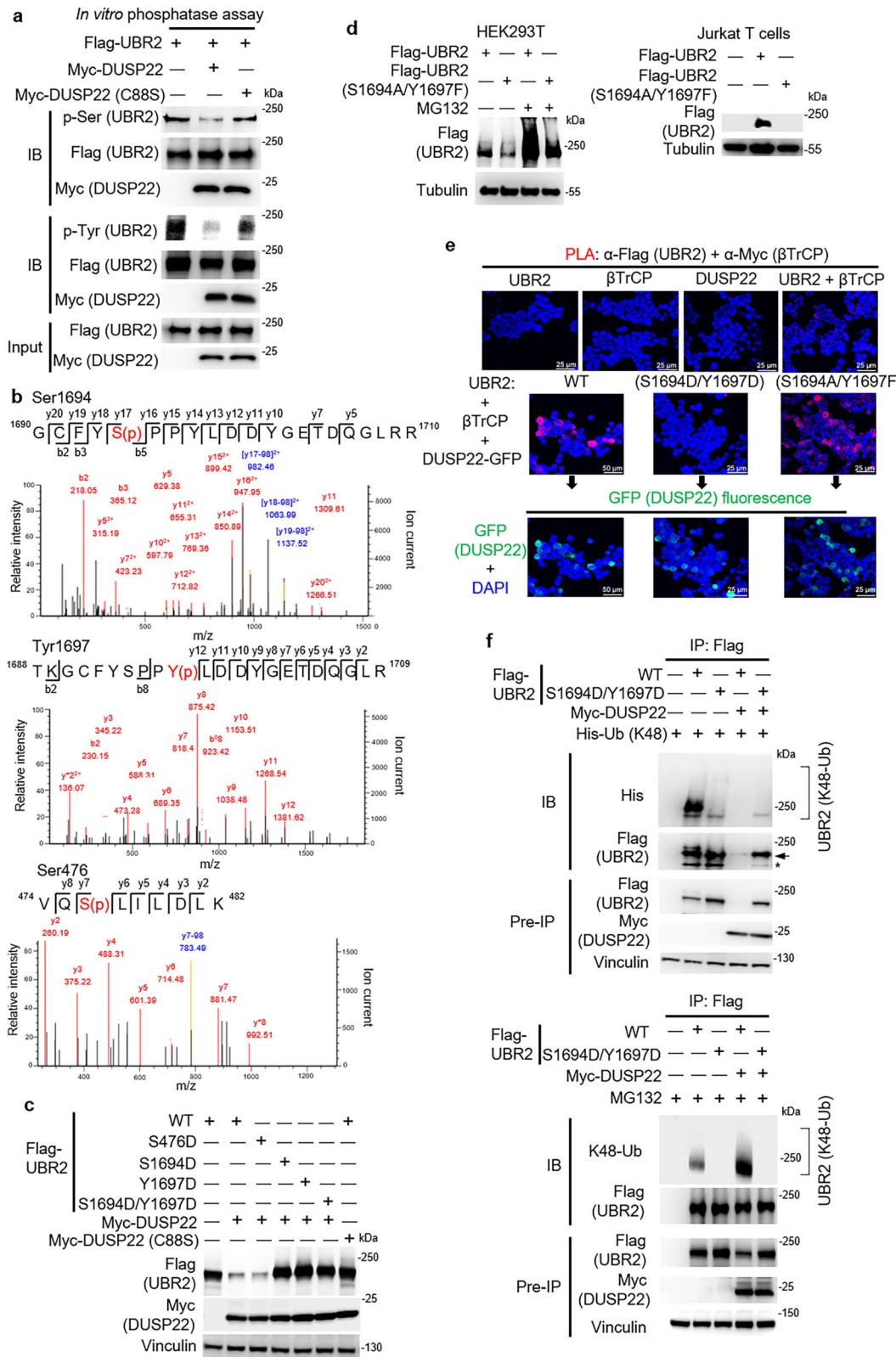

To verify that these identified lysine residues are targets for Lys63-linked ubiquitination by UBR2, Lys99 and Lys276 residues were individually mutated to arginine residues. Immunoprecipitation and Western blotting analysis showed that individual Lck (K99R) mutation, Lck (K276R) mutation, and Lck (K99/276R) double mutations significantly alleviated UBR2-induced Lys63-linked ubiquitination of Lck in HEK293T cells (Fig. 6f). Notably, the interaction between Lck and

UBR2 proteins was not affected by ubiquitination deficiency (K99/276R double mutations) of Lck (Supplementary Fig. 8a), suggesting that K99/276 R double mutations do not cause a globally conformational change of Lck. These results suggest that UBR2 ubiquitinates Lck at Lys99 and Lys276 residues.

To further study whether mutations of the Lys63-linked ubiquitination sites of Lck inhibit Lck Tyr394 phosphorylation during TCR

**Fig. 4 | DUSP22 induces UBR2 degradation by dephosphorylating Ser1694 and Tyr1697 residues of UBR2. a** DUSP22 directly dephosphorylated UBR2. In vitro phosphatase assays were performed using Myc-tagged DUSP22 wild-type or phosphatase-dead mutant (C88S) immunocomplex incubated with Flag-tagged UBR2 immunocomplex (substrates). The reactant products were subjected to immunoblotting analyses. **b** The Ser1694, Tyr1697, and Ser476 residues were identified as the DUSP22-targeted dephosphorylation sites on UBR2. The MS/MS fragmentation spectra of the tryptic peptides of UBR2 contain the phosphorylation modifications. S(p) and Y(p) denote phosphorylated serine and tyrosine residues, respectively. The blue color denotes the peptides corresponding to the loss of phosphate group from these peptides. **c** The UBR2 phosphomimetic mutants (S1694D, Y1697D, or S1694D/Y1697D) were resistant to the DUSP22-induced degradation. Myc-DUSP22 plus individual Flag-UBR2 mutants (S476D, S1694D, Y1697D, or S1694D/Y1697D) plasmids were co-transfected into HEK293T cells. The levels of Flag-tagged UBR2 proteins were determined by immunoblotting analysis. Vinculin immunoblotting was performed by reprobing the Flag (UBR2) immuno-blot membrane with anti-vinculin antibody. **d** The phosphorylation-deficient mutation of UBR2 (S1694A/Y1697F) decreased protein stability. Flag-UBR2 or Flag-

UBR2 mutant (S1694A/Y1697F) plasmid was transfected into HEK293T (with or without 25 μM MG132, 4 h) or Jurkat T cells. The cell lysates were subjected to immunoblotting analysis using anti-Flag antibody. **e** Phosphomimetic double mutations (S1694D/Y1697D) of UBR2 blocked the DUSP22-induced UBR2–βTrCP interaction. Myc-βTrCP, GFP-DUSP22, and individual Flag-UBR2, Flag-UBR2 (S1694D/Y1697D), and Flag-UBR2 (S1694A/Y1697F) plasmids were co-transfected into HEK293T cells. Red fluorescence represents the interactions ( < 40 nm) of UBR2 with βTrCP. GFP fluorescence (green color) indicates the expression of GFP-DUSP22. Images were captured with 400X original magnification by confocal microscope (Leica TCS SP5 I). Cell nuclei were stained with DAPI. Scale bar, 25 or 50 μm. **f** Phosphomimetic mutation (S1694D/Y1697D) of UBR2 reduced UBR2 Lys48-linked ubiquitination. Myc-DUSP22 and His-Ub (K48) plus either Flag-UBR2 or Flag-UBR2 mutant (S1694D/Y1697D) plasmids were co-transfected into HEK293T cells. Cells were treated with or without 25 μM MG132 for 4 h. UBR2 wild-type (WT) or phosphomimetic mutant (S1694D/Y1697D) proteins were immunoprecipitated with anti-Flag antibody and then immunoblotted with anti-His (top panel), anti-ubiquitin (Lys48) (bottom panel), or anti-Flag antibody. Arrow, intact UBR2 protein; asterisk, degraded UBR2.

signaling, immunoprecipitated Flag-tagged Lck proteins from Jurkat T cells were subjected to Western blotting analysis. TCR signaling stimulated Tyr394 phosphorylation of Lck (Fig. 6g), whereas the TCR-induced phosphorylation of Lck were abolished by Lck (K99/276R) double mutations in Jurkat T cells (Fig. 6g). Moreover, PLA data showed that TCR signaling-induced Lys63-linked ubiquitination and Tyr394 phosphorylation of Lck in Jurkat T cells transfected with Flag-Lck plasmid were abolished by Lck (K99/276R) double mutations (Fig. 6h, and Supplementary Fig. 8b, c). These results suggest that Lys99 and Lys276 residues are required for Lys63-linked ubiquitination and Tyr394 phosphorylation of Lck in anti-CD3-stimulated T cells. To examine whether Lck (K99/276R) double mutant inhibits Lck trans-autophosphorylation, an in vitro kinase assay was performed using purified Flag-tagged Lck wild-type, Lck (K99/276R), or Lck (Y394F) mutant proteins from Jurkat T cells co-transfected with Myc-UBR2 plasmid. Additional Flag-tagged Lck proteins isolated from HEK293T cells without any stimulation were used as substrates. The wild-type Lck (from Jurkat T cells) induced Tyr394 phosphorylation of Lck (from HEK293T cells) (Fig. 6i), which was not induced by Lck ubiquitination-deficient (K99/276R) mutant or Lck (Y394) phosphorylation-deficient mutant (Fig. 6i). These results suggest that Lys63-linked ubiquitination of Lck at Lys99 and Lys276 residues induces trans-autophosphorylation and kinase activity of Lck. To examine whether trans-autophosphorylation of Lck at Tyr394 residue is induced by UBR2-mediated Lys63-linked ubiquitination, we performed in vitro ubiquitin E3 ligase assay in combination with in vitro kinase assay. Purified Flag-tagged Lck or Flag-tagged Lck ubiquitination-deficient (K99/276R) mutant proteins were coincubated with Myc-tagged UBR2 proteins plus E1 (UBE1), E2 (UBE2N), and His-ubiquitin proteins. After in vitro ubiquitin E3 ligase assay, the kinase buffer was added for in vitro kinase assay. We found that trans-autophosphorylation of wild-type Lck was induced by UBR2-mediated Lys63-linked ubiquitination, whereas the trans-autophosphorylation of Lck ubiquitination-deficient (K99/276R) mutant was undetectable (Fig. 6j). These results suggest that UBR2-induced Lys63-linked ubiquitination of Lck at Lys99 and Lys276 residues are required for Lck Tyr394 trans-autophosphorylation.

### DUSP22 inhibits T cell-mediated inflammation by down-regulating UBR2 functions

DUSP22 plays a negative regulator in T-cell activation and inflammation by dephosphorylating Lck[14]; conversely, DUSP22 knockout mice display spontaneous autoimmune phenotypes[14]. To study whether that the induction of inflammation in DUSP22 knockout mice is due to UBR2-mediated Lck activation, we bred DUSP22

knockout mice with UBR2 knockout mice to generate DUSP22/UBR2 double knockout mice. DUSP22 knockout and DUSP22/UBR2 double knockout were confirmed by immunoblotting analyses (Supplementary Fig. 8d). Tyr394 phosphorylation of Lck was induced by TCR signaling in T cells, while the Lck phosphorylation was further enhanced by DUSP22 knockout (Fig. 7a). The enhanced Lck Tyr394 phosphorylation by DUSP22 knockout was abolished by UBR2 knockout in T cells (Fig. 7a), suggesting that DUSP22 inhibits UBR2-induced Lck Tyr394 phosphorylation in TCR signaling. Moreover, PLA data showed that TCR-induced Lys63-linked ubiquitination and Tyr394 phosphorylation of Lck were drastically enhanced by DUSP22 knockout, whereas the DUSP22 deficiency-induced ubiquitination and phosphorylation of Lck was abolished by UBR2 knockout (Fig. 7b, c). These results support that the induction of Lck Lys63-linked ubiquitination and Tyr394 phosphorylation in DUSP22 knockout cell is due to UBR2 overexpression. To further demonstrate the role of UBR2 in promoting inflammatory responses of DUSP22 knockout mice, serum cytokines from DUSP22/UBR2 double knockout and control mice (wild-type and DUSP22 knockout mice) were determined by ELISA assay. The induction of serum IFN-γ, TNF-α, and IL-17A levels in DUSP22 knockout mice were significantly decreased by UBR2 knockout (Fig. 7d). Moreover, histology staining showed kidney glomerulomegaly, inflamed lungs, and inflamed livers were drastically ameliorated in aged DUSP22/UBR2 double-knockout mice compared to those of aged DUSP22 knockout mice (Fig. 7e). The results suggest that inflammation in DUSP22 knockout mice is mediated by UBR2. Furthermore, we performed Western blotting of UBR2, p-Lck, DUSP22, and Lck proteins using T cells of SLE patients. We found that UBR2 and p-Lck protein levels were upregulated, whereas DUSP22 protein levels were decreased, in T cells of SLE patients (Fig. 7f). We next studied whether the UBR2-Lck interaction and Lck Lys63-linked ubiquitination are induced in T cells of human SLE patients. Peripheral blood T cells from human SLE patients displayed PLA signals of the UBR2-Lck interaction, whereas T cells from healthy controls did not (Fig. 7g). Lys63-linked ubiquitination of Lck was also induced in SLE patients compared to that of healthy controls (Fig. 7h), suggesting that UBR2 induces Lys63-linked ubiquitination of Lck in human SLE T cells (Fig. 7g, h). Collectively, these results suggest that DUSP22 inhibits ubiquitination/phosphorylation of Lck by downregulating UBR2, contributing to suppression of T-cell activation and T cell-mediated autoimmune responses. To study whether DUSP22 inhibits T-cell activation through UBR2 degradation in a T cell-intrinsic manner, antigen-specific T-cell-mediated cytokine production was examined using T cells from mice with experimental autoimmune encephalomyelitis (EAE). Female mice are required for optimal induction of

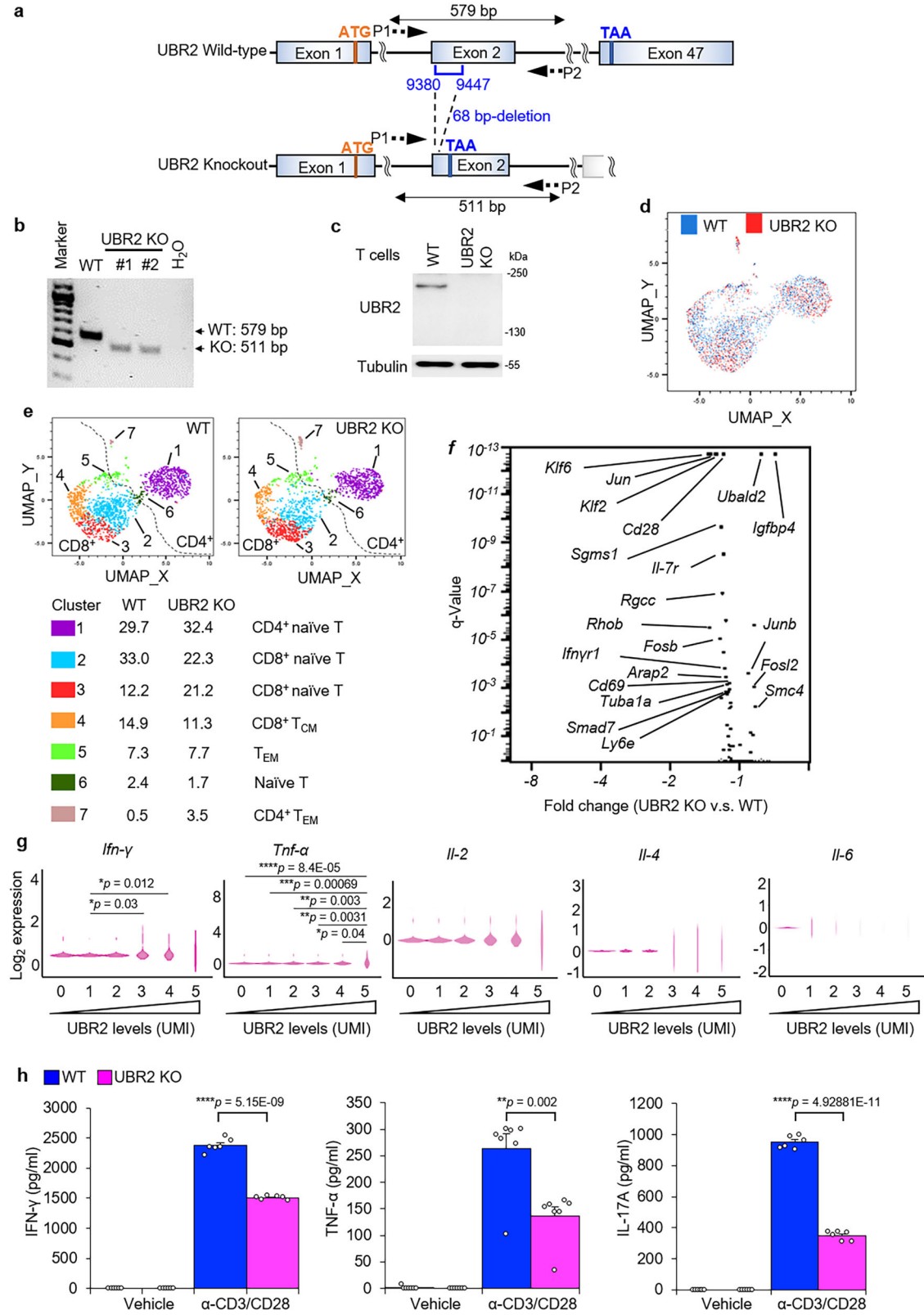

EAE[35]. Due to the lack of female DUSP22/UBR2 double knockout mice, we characterized the T cells isolated from lymph nodes of female DUSP22/UBR2 heterozygous knockout mice. IFN-γ, TNF-α, and IL-17 levels in MOG-restimulated DUSP22 knockout T cells were enhanced compared to those of wild-type T cells (Fig. 7i). In contrast, the induction of the proinflammatory cytokines in DUSP22 knockout T cells was abolished by UBR2 heterozygous knockout

(Fig. 7i), suggesting that the attenuated autoimmune phenotype observed in DUSP22/UBR2 heterozygous knockout (DUSP22[−/−]; UBR2[+/−]) mice is due to suppression of hyperactive T cells, but may also involve other cell types. Taken together, UBR2-induced Lck trans-autophosphorylation and Lys63-linked ubiquitination were further enhanced by DUSP22 deficiency, leading to proinflammatory cytokines production and autoimmune disease.

**Fig. 5 | scRNA-seq analysis of T cells reveals that UBR2 stimulates cytokine production. a** Schematic diagram of the murine UBR2 wild-type (WT) alleles and the targeted UBR2 mutant alleles. P1 and P2, the primers for PCR. **b** PCR analyses of wild-type and UBR2 mutant alleles using genomic DNAs from mouse tails. The PCR product of the upper band (579 bp) denotes the wild-type allele, and the lower band (511 bp) denotes the UBR2 mutant allele. **c** Immunoblotting analyses of UBR2 protein levels in T cells of wild-type or UBR2 knockout (KO) mice. **d** UMAP plot showed dimensional reduction of the distribution of 3061 cells. T cells derived from each group are shown in different colors (blue: wild-type; red: UBR2 knockout). **e** UMAP plot showed seven major clusters of the 3061 individual T cells. The number denotes the cluster frequency in parent population. **f** Volcano plot showed the differential expression transcripts between UBR2 knockout and wild-type T cells. Fold change represents the gene expression in UBR2 knockout T cells versus wild-type T cells. **g** Violin plots showed the expression levels of IFN-γ, TNF-α, IL-2, IL-4, and IL-6 genes in wild-type T cells under different UBR2 expression levels. The mRNA levels of UBR2 and cytokines were detected as unique molecular identifier (UMI) counts. **h** ELISAs of IFN-γ, TNF-α, and IL-17A levels in culture supernatants from peripheral blood T cells treated with plate-bound anti-CD3 plus anti-CD28 costimulation for 72 h. Total murine primary T cells were used in Fig. 5c-h.

## Discussion

A key finding of our study is the identification of UBR2 as an E3 ligase that induces ubiquitination and activation of Lck in TCR signaling. UBR2 induced Lck Lys63-linked ubiquitination at Lys99 and Lys276 residues upon TCR stimulation. The UBR2-induced Lys63-linked ubiquitination of Lck in T cells was required for Lck Tyr394 phosphorylation, controlling Lck activation and subsequent T-cell activation. Conversely, TCR-stimulated proinflammatory cytokine production in murine T cells was attenuated by UBR2 knockout. Consistently, the UBR2-Lck interaction and Lck Lys63-linked ubiquitination were induced in T cells of human SLE patients. Our study provides the insight that UBR2 is the critical upstream activator of Lck in TCR signaling and T-cell-mediated inflammation.

Mutations of β3/α loop of chicken Src at Pro299 and Pro304 residues results in a constitutively active form of Src even in the presence of CSK[36,37]. The residues of chicken Src at Pro299 and Pro304 residues were aligned to human Lck Pro277 and Gln282 residues, which were adjacent to the K63-ubiquitination site (Lys276) in kinase domain of human Lck. The linker of Lck connects the catalytic domain with the SH2 domain and SH3 domain[36]. Another ubiquitination site (Lys99) of human Lck was located adjacent to Trp97 and Phe110 residues in SH3 catalytic domain of Lck; Trp97 and Phe110 residues interact with linker and stabilize the closed conformation of Lck[36]. It is likely that the conjugation of Lys63-linked ubiquitin proteins at Lys99 and Lys276 of Lck may result in disruption of the closed conformational state of Lck, leading to an open conformation of Lck, providing an opportunity for Tyr394 trans-autophosphorylation of Lck.

Lck kinase was thought to be in a constitutively active state in resting T cells, independent of TCR and coreceptor[38]. The basal levels of Lck Tyr394 autophosphorylation may be maintained by high Lck protein levels[38]. In contrast, TCR-induced de novo phosphorylation of Lck at Tyr394 residue together with the conformational opening of Lck are essential for Lck-induced ITAM phosphorylation and the initiation of T-cell activation[39]. Our finding provides an understanding on the initial steps of T cell receptor signaling and the conformational changes in Lck. In this study, we propose the concept of Lys63-linked ubiquitination as a novel mechanism for activating Lck. Our results suggest that UBR2-induced Lys63-linked ubiquitination plays a crucial role in the initiation of Lck Tyr394 phosphorylation and subsequent activation. The Lys63-linked ubiquitination of Lck may disrupt the closed conformation of Lck, facilitating autophosphorylation at Tyr394. This mechanistic insight underscores the significance of K63-ubiquitination-mediated conformational changes in initiating Lck activation and enables understanding Lck regulation in T-cell receptor signaling.

Another exciting finding in this study is that DUSP22 inhibits T-cell activation by inducing UBR2 degradation. After UBR2-mediated Lck activation in the TCR signaling turn-on stage, DUSP22 interacted with and downregulated UBR2 in activated T cells. In the TCR turn-off stage, DUSP22 dephosphorylated UBR2 at Ser1694 and Tyr1697 residues, leading to SKP1-CUL1-βTrCP complex-induced UBR2 degradation. DUSP22 is critical for suppression of T-cell-mediated inflammation[14]; DUSP22 downregulation in T cells causes systemic inflammation in mice[10]. UBR2 upregulation is associated with autoimmune pancreatitis and chronic lymphocytic leukemia[22]. Our study showed that UBR2 levels were not inducible by α-CD3 stimulation during the TCR signaling turn-on stage. Notably, DUSP22 depletion resulted in UBR2 upregulation in T cells, leading to enhancement of proinflammatory cytokine production. The induction of proinflammatory cytokine production by DUSP22 knockout were blocked by UBR2 knockout. Furthermore, DUSP22 downregulation and UBR2 upregulation occurred in T cells of human SLE patients. These findings suggest that DUSP22 downregulation causes UBR2 overproduction in T cells, resulting in inflammatory diseases and immune-dysregulation disorders.

In conclusion, in TCR signaling turn-on (early activation) stage, UBR2 plays a stimulatory role in the activation of Lck via Lys63-linked ubiquitination of Lck, leading to enhanced T-cell activation (Fig. 8a). In contrast, DUSP22 can inhibit Lck in TCR signaling turn-off stage indirectly through pathway I and directly through pathway II. In pathway I, DUSP22 dephosphorylates UBR2, resulting in the SKP1-CUL1-βTrCP complex-induced UBR2 degradation and subsequent inhibition of UBR2-mediated Lck activation (Fig. 8b). In pathway II, DUSP22 directly dephosphorylates the activated Lck, leading to the obliteration of Lck activation[14] (Fig. 8b). Thus, our mechanistic insights on Lck activation and inactivation by UBR2 and DUSP22, respectively, may provide therapeutic strategies to reduce T-cell-mediated autoimmune disease.

## Methods
### Human participants
This study was conducted in accordance with the Helsinki Declaration. For collecting peripheral blood samples, 6 health individuals and 6 SLE patients who underwent clinic were invited to participate in academic research by providing peripheral blood samples. The selection of SLE patients was based on a well-defined SLE diagnosis. The peripheral blood collection, T-cell purification, and the experiments were approved by the ethics committees of Taipei Veterans General Hospital (2017-06-003BC) and National Taiwan University Hospital (109-008-E). All study participants provided written informed consent prior to enrollment. For PLA assay, a total of 6 individuals, including 3 healthy individuals and 3 SLE patients were enrolled in this study from Division of Immunology and Rheumatology at Taipei Veterans General Hospital, Taiwan. For immunoblotting, 3 healthy individuals were enrolled from National Taiwan University Hospital and 3 SLE patients were enrolled from the Division of Immunology and Rheumatology at Taipei Veterans General Hospital.

### Mice
All mouse experiments were performed in the Association for Assessment and Accreditation of Laboratory Animal Care International (AAALAC)–accredited animal housing facilities at the National Health Research Institutes (NHRI). All mice were used according to the protocols and guidelines approved by the Institutional Animal Care and Use Committee of NHRI. DUSP22 knockout mice were generated by TALEN-mediated gene targeting using pronuclear microinjection by Transgenic Mouse Core of NHRI. The deletion of 4 bp (85GATC88) in exon 1 resulted in a 24-amino acid frame-shift mutant

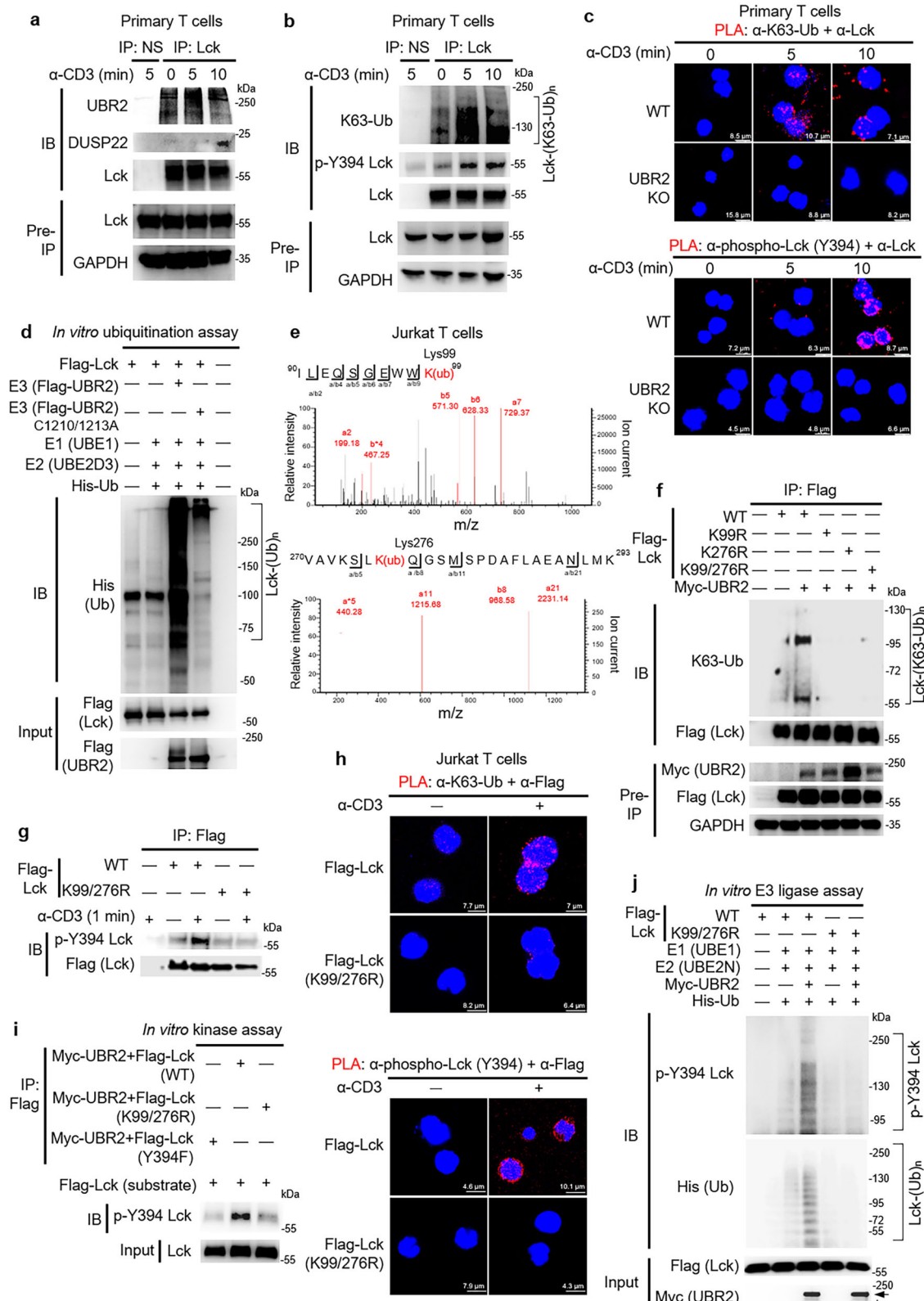

(MGSGMSHCRACTLATSKTQEMQNSstop) of DUSP22. The final 18 amino acids (HCRACTLATSKTQEMQNS) were generated due to frameshift mutation. UBR2 knockout mice were generated by CRISPR-mediated gene targeting using pronuclear microinjection by Transgenic Mouse Core of NHRI. The deletion of 68 bp ([9380]GCTGCAAGCAACCGACCTCAACAGAGAAGTGTACCAGCATTTAG-CCCACTGTGTGCCCAAAATCTACT[9447]) in exon 2 resulted in a 31-amino

acid frame-shift mutant (MASEMEPEVQAIDRSLLECSAEEIAGRWPGP-stop) of UBR2. The final 3 amino acids (PGP) were generated due to frame-shift mutation. Sex-matched, 12- to 24-week-old C57BL/6 J wild-type, UBR2 knockout, DUSP22 knockout, DUSP22/UBR2 double knockout, and DUSP22[−/−];UBR2[+/-] mice were used in this study. Mice were bred and housed at the specific-pathogen-free cages with at most 4 animals per cage. These mice were maintained in a 12 h light/12 h dark

**Fig. 6 | UBR2 induces Lck activation by K63-ubiquitinating Lys99 and Lys276 residues of Lck. a** Immunoprecipitations of the endogenous Lck with either UBR2 or DUSP22 proteins in the lysates of anti-CD3-stimulated murine primary T cells. **b** TCR signaling stimulated Lys63-linked ubiquitination and Tyr394 phosphorylation of Lck in murine primary T cells. Endogenous Lck immunocomplexes in the lysates of anti-CD3-stimulated T cells were immunoprecipitated with anti-Lck antibody and then subjected to immunoblotting. **c** Confocal microscopy analyses of PLA for the ubiquitinated and phosphorylated Lck in TCR-stimulated T cells of wild-type (WT) or UBR2 knockout mice using anti-Lck antibody plus either anti-ubiquitin (Lys63) or anti-phospho-Lck (Tyr394) antibodies. Red fluorescence represents the endogenous Lck proteins containing Lys63-linked ubiquitination or Tyr394 phosphorylation. Cell nuclei were stained with DAPI. **d** UBR2 induced Lck ubiquitination in vitro. Recombinant His-ubiquitin, E1 (UBE1), E2 (UBE2D3), Flag-Lck, and ATP were co-incubated in the ubiquitination buffer with E3 (Flag-tagged UBR2 or Flag-tagged UBR2 ligase-inactive mutant (C1210/1213A)) proteins. **e** The Lys99 and Lys276 residues were identified as ubiquitination sites of Lck by mass spectrometry analysis. K(ub), ubiquitinated lysine residue. **f** Immunoprecipitations of ubiquitinated Flag-tagged Lck with Myc-tagged UBR2 proteins in the lysates of indicated HEK293T transfectants. Flag-tagged Lck proteins were immunoprecipitated with anti-Flag antibody and then immunoblotted with anti-ubiquitin (Lys63) antibody or anti-Flag antibody. **g** Immunoprecipitation and immunoblotting analysis of Flag-tagged Lck with Tyr394 phosphorylated Lck proteins in the lysates of Jurkat T cells transfected with Flag-Lck (WT) or Flag-Lck (K99/276R) plasmid. **h** Confocal microscopy analyses of PLA for the ubiquitinated Lck and Tyr394-phosphorylated Lck in TCR-stimulated Jurkat transfectants using anti-Lck plus anti-ubiquitin (Lys63) and anti-phospho-Lck (Tyr394) antibodies, respectively. Red fluorescence represents the endogenous Lck proteins containing Lys63-linked ubiquitination or Tyr394 phosphorylation. Cell nuclei were stained with DAPI. **i** Double mutations (K99/276R) of Lck inhibited phosphorylation of Lck at Tyr394 residue by in vitro kinase assay. **j** In vitro ubiquitin E3 ligase assay in combination with kinase assay showed that ubiquitination-deficient Lck inhibited Lys63-linked polyubiquitination-induced trans-autophosphorylation of Lck by UBR2. Arrow, intact UBR2 protein; asterisk, degraded UBR2 protein.

cycle, and the housing temperature and humidity were maintained at 24 °C and 50%, respectively. All mice were euthanized by $CO_2$ inhalation.

## Plasmids and antibodies

Flag-UBR2 wild-type plasmid was constructed by subcloning UBR2 cDNA into the vector pCMV6-AC-Flag (OriGene Technologies). The pCMV6-UBR2 (S1694D, Y1697D, S1694D/Y1697D, S1694A/Y1697F, C1210/1213A, or Lys-to-Arg mutations) and pCMV6-Lck (K99/276R) mutant plasmids were generated by PCR mutagenesis. Flag-Lck, Myc-DUSP22, Myc-DUSP22, Myc-DUSP22 (C88S), GFP-DUSP22, GFP-DUSP22 (C88S), and GST-DUSP22 were described previously[14,40]. Myc-CUL1, Myc-SKP2, Myc-FBXW7, and Myc-βTrCP plasmids were constructed by cloning the cDNA into vector pCMV-3Tag-9 (Agilent Technologies). Flag-CUL1 and Flag-βTrCP plasmids were purchased from OriGene. Lys11-only ubiquitin (K11-Ub) plasmid was purchased from Addgene (#22901). The DUSP22, CUL1, and βTrCP short hairpin RNA (shRNA) constructs were purchased from the National RNAi Core Facility (Taiwan); DUSP22 shRNA #1 and #2 target sequences are 5'-TACCTGTGCATCC-CAGCAG-3' and 5'-ACACTGGTGATCGCATACA-3'. CUL1 shRNA #1, #2, and #3 target sequences are 5'- GCCAGCATGATCTCCAAGTTA-3', 5'-CCCGCAGCAAATAGTTCATGT-3', and 5'- GCACACAAGATGAATTA GCAA-3', respectively. βTrCP shRNA #1 and #2 target sequences are 5'-GCGTTGTATTCGATTTGATAA-3' and 5'-GCTGAACTTGTGTGCAAG-GAA-3'. Anti-Flag (clone M2, Cat F3165, Lot SLCJ3741), anti-Myc (clone 9E10, Cat M4439), anti-Lys48-specific ubiquitin (clone Apu2, Cat 05-1307, Lot ZRB2150), anti-Lys63-specific ubiquitin (clone HWA4C4, Cat 05-1313, Lot 3147606), anti-vinculin (clone VIIF9, Cat MAB3574, Lot 3951156), anti-GST (clone DG122-2A7, Cat 05-311), and anti-His (clone 6AT18, Cat SAB1305538, Lot SA141112AB) monoclonal antibodies were purchased from Sigma-Aldrich. Anti-CUL1 (clone D5, Cat sc-17775, Lot B0714), and anti-CD3 zeta (clone 6B10.2, Cat sc-1239, Lot J1507) antibodies were purchased from Santa Cruze Biotechnology. Anti-UBR2 (Cat PA5-37161, Lot VG3023905) and anti-tubulin (clone BT7R, Cat MA5-16308, Lot QG218956) antibodies were purchased from Thermo Fisher Scientific. Anti-HA (clone C29F4, Cat 3724 S, Lot 8), anti-phospho-ZAP70 (Y319) (Cat 2701, Lot 7), anti-phospho-ZAP70 (Y493) (Cat 2704, Lot 3), anti-ZAP70 (clone 99F2, Cat 2705, Lot 2), anti-phospho-LAT (Y171) (Cat 3581, Lot 1), anti-phospho-PLCγ1 (Y783) (Cat 2821, Lot 7), anti-βTrCP (clone D13F10, Cat 4394, Lot 4), and anti-Lck (Cat 2752, Lot 5) antibodies were purchased from Cell Signaling. Anti-GAPDH (clone mAbcam 9484, Cat ab9484, Lot 39238413-1) antibody was purchased from Abcam. Anti-phospho-Lck (Y394) monoclonal antibody (clone 7551-3, Cat MAB7500, Lot CHAZ0112071) was purchased from Bio-Techne. A homemade anti-DUSP22 antibody (DUSP22-C) was generated by immunization of mice with peptides (murine DUSP22 epitope: [181]RPSSRRWSSFSTLPPLTYNNYTTET[205]).

Homemade anti-DUSP22 monoclonal antibody (clone C8) was generated by immunization of mice with peptides (murine DUSP22 epitope: [170]GKYKEQGRTEPQPGARRWSS[189])[11]. Homemade anti-phospho-ERK1/2 (T202/Y204) antibody was generated by immunization of mice with peptides (murine p-ERK epitope: [133]HTGFLTEpYVATRW[145]). Homemade anti-ERK1/2 (T202/Y204) antibody was generated by immunization of mice with peptides (murine ERK epitope: [308]HPYLEQYYDPSDEPIAEAP[326]). Anti-CD3ζ (Y142) antibody (Cat A02421Y142, Lot A02421Y142) was purchased from Boster Biotechnology. Anti-phospho-LAT (Y191) (Cat 07-278, Lot 22138) and anti-LAT (clone 11B.12, Cat 05-770, Lot 26805) antibodies were purchased from EMD Millipore. Anti-PLCγ1 monoclonal antibody (clone B-6-4, Cat 05-366, Lot 23344) was purchased from Merck. For immunoblots, all primary antibodies were used at a 1:1,000 dilution, except for anti-Myc antibody (1:100,000). For PLA assays, all primary antibodies were used at a 1:300 dilution, except for anti-Flag and anti-Myc antibodies (1:1,000). The antibodies for flow cytometry—including anti-mCD3ε-FITC (clone 145-2C11, Cat 100306, Lot B151259), anti-F4/80-PE/Cy7 (clone BM8, Cat 123114, Lot B207313), and anti-IgM-APC (clone RMM-1, Cat 406509, Lot B197034) antibodies—were purchased from BioLegend. Anti-mCD4-pacific blue (clone RM4-5, Cat 558107, Lot 2202191), anti-CD8-APC-Cy7 (clone 53-6.7, Cat 557654, Lot 1008430), anti-CD11b-FITC (clone M1/70, Cat 553310, Lot 74545), and anti-CD45R/B220-PE/Cy7 (clone RA3-6B2, Cat 552772, Lot B144210) antibodies were purchased from BD Biosciences. Anti-Foxp3-FITC monoclonal antibody (clone FJK-16s, Cat 53-5773-82, Lot 2416220) was purchased from Invitrogen. All antibodies for flow cytometry were used at 1:50 or 1:100 dilutions. The antibodies for T cell activation: anti-CD3 (clone OKT3, Cat 317315, Lot B144612) antibody was purchased from BioLegend, anti-CD3 (clone 145-2C11, Cat 553057, Lot 1116065), anti-CD3 (clone 500A2, Cat 553239, Lot 0314175), and anti-CD28 (clone 37.51, Cat 553294, Lot 1039452) antibodies were purchased from BD Pharmingen.

## Cell lines and transfection

HEK293T cells (American Type Cell Culture, CRL-11268) were cultured in DMEM medium (Invitrogen) containing 10% FBS plus penicillin and streptomycin (Invitrogen). HEK293T cells were transfected using polyethylenimine (PEI) reagents for 48 h. For the treatment of inhibitors/reagents, HEK293T cells were treated with 25 μM MG132 (Calbiochem) for 4 h, 50 μM Z-VAD-FMK (R&D Systems), or 50 μM chloroquine (Sigma) for the indicated time points. Human Jurkat (E6-1 clone)[41] (American Type Cell Culture, TIB-152) T leukemia cells, their derivative J.Cam1.6 clone[42] (Lck deficient) (American Type Cell Culture, CRL-2063), and J-TAg clone cells[14], were cultured in RPMI-1640 medium (Invitrogen) containing 10% fetal bovine serum (FBS) plus penicillin and streptomycin (Invitrogen). Jurkat T cells were transfected by

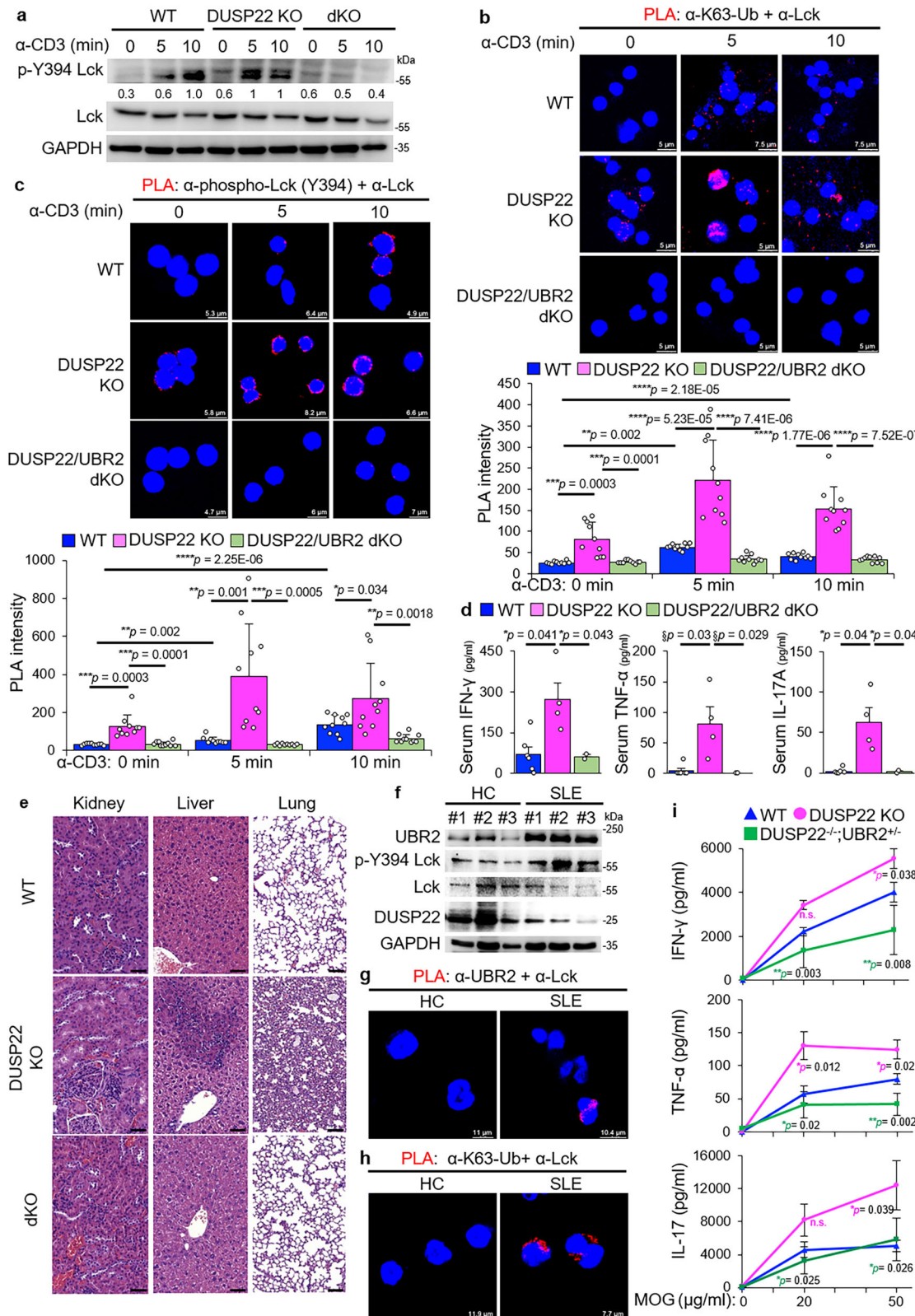

electroporation method using the Neon Transfection System (Invitrogen). Human Jurkat (J-TAg clone) T cells were used in most of T cell-line experiments, except for mass spectrometry assay (Fig. 6e).

### Coimmunoprecipitation and immunoblotting analysis

To extract protein lysates, cells were lysed in RIPA buffer (1% sodium deoxycholate, 1 mM EDTA, 150 mM NaCl, 1% Triton X-100, 0.1% SDS,

50 mM Tris-HCl) supplemented with 0.5% protease inhibitor cocktail (#539134, EMD Millipore), 1% phosphatase inhibitor cocktail (#524629, EMD Millipore), and 1 mM $Na_3VO_4$ at 4 °C for 30 min. For immuno-precipitation, 1 mg protein lysates were incubated with anti-Myc agarose beads (clone 9E10, Sigma-Aldrich) or anti-Flag agarose beads (clone M2, Sigma-Aldrich) in 1 ml of lysis buffer at 4 °C for 1-2 h. Primary T cell lysates were incubated with individual antibodies and

**Fig. 7 | DUSP22 deficiency results in T-cell-mediated inflammation through UBR2. a** Immunoblotting of phospho-Tyr394 Lck, Lck, and GAPDH proteins from purified peripheral blood murine T cells of wild-type (WT), DUSP22 knockout (KO), or DUSP22/UBR2 double knockout (dKO) mice upon anti-CD3 stimulation. Quantification of p-Lck proteins (normalized to Lck proteins) is shown at the bottom of p-Lck panel. **b, c** Confocal microscopy analyses of PLA for ubiquitinated Lck (**b**) and Tyr394 phosphorylated Lck (**c**) proteins in TCR-stimulated T cells of wild-type, DUSP22 knockout, or DUSP22/UBR2 double knockout (dKO) mice using anti-Lck antibody plus anti-ubiquitin (Lys63) antibody (**b**) and anti-phospho-Lck (Tyr394) antibody (**c**), respectively. Red fluorescence represents the endogenous Lck proteins containing Lys63-linked ubiquitination or Tyr394 phosphorylation. Cell nuclei were stained with DAPI. The red signal intensity was plotted in the lower panels. **d** Serum levels of IFN-γ, TNF-α, and IL-17A in wild-type, DUSP22 knockout (KO), and DUSP22/UBR2 double knockout (dKO) mice were determined by ELISAs. **e** Hematoxylin and eosin (H&E)-stained sections of the kidney, liver, and lung from 10- to 11-month-old wild-type, DUSP22 knockout (KO) and DUSP22/UBR2 double knockout (dKO) mice. Scale bar, 50 μm (kidney and liver) and 100 μm (lung). **f** Western blotting analysis of UBR2, p-Lck, and DUSP22 proteins in peripheral blood T cells of human SLE patients and healthy controls (HC). Anti-GAPDH immunoblotting was performed by reprobing the anti-Lck immunoblot membrane. **g, h** Confocal microscopy analyses of PLA for the interaction between the endogenous UBR2 and Lck proteins or Lys63-linked ubiquitination of Lck in peripheral blood T cells of a representative human SLE patient and a representative healthy control (HC). T-cell nuclei were stained with DAPI. **i** ELISAs of IFN-γ, TNF-α, and IL-17A levels in culture supernatants from MOG-restimulated T cells. Mice were immunized with MOG peptide emulsified in CFA, followed by injection of pertussis at day 0, 1, and 2. To determine the activation of Ag-specific T cells, T cells from lymph nodes of the immunized mice were cultured in the presence of 0, 20, and 50 μg/ml MOG for 72 h, and cytokine levels of IFN-γ, TNF-α, and IL-17A were analyzed by ELISA. n.s., not significant.

protein A/G-agarose beads (GE17-1279-01/GE17-0618-01, Merck) in lysis buffer at 4 °C for 2 h. The immunocomplexes were washed three times with lysis buffer, followed by immunoblotting analysis. For immunoblotting analysis, cell lysates or immunocomplexes were boiled at 95 °C for 5 min. Protein lysates or immunocomplexes were loaded on SDS-PAGE, and the separated proteins were transferred to PVDF membranes. The PVDF membranes were blocked with 5% BSA in TBST for 2 h and then incubated with specific primary antibodies at 4 °C overnight, followed by incubation with corresponding secondary antibodies at room temperature for 1 h. The specific signals were detected by ECL substrate kit (Millipore). For immunoblotting, signals were detected by Syngene and analyzed by Pxi9 Access GENESys software.

## Protein denaturation and re-immunoprecipitation

To determine UBR2 ubiquitination, HEK293T cells were transfected with Flag-UBR2 plus Myc-DUSP22 and treated with MG132 (25 μM). UBR2 was immunoprecipitated with anti-Flag antibody (1st IP); half of the first anti-Flag immunoprecipitates were boiled in the presence of 1% SDS, renatured by serial dilution with RIPA buffer (1% sodium deoxycholate, 1 mM EDTA, 150 mM NaCl, 1% Triton X-100, 0.1% SDS, 50 mM Tris-HCl), followed by a second round of immunoprecipitation (2nd IP) with anti-Flag antibody. UBR2 ubiquitination was determined by immunoblotting analyses using anti-ubiquitin antibody.

## Liquid chromatography-mass spectrometry (LC-MS/MS)

For the identification of DUSP22-interacting proteins, immunocomplexes of Myc-tagged DUSP22 were immunoprecipitated with anti-Myc agarose beads from lysates of HEK293T cells transfected with vector or Myc-DUSP22 plasmid. For identification of DUSP22-induced UBR2 Lys48-linked ubiquitination residues, immunocomplexes of Flag-tagged UBR2 were immunoprecipitated with anti-Flag antibody from lysates of HEK293T cells co-transfected with Flag-UBR2 and Lys48-only ubiquitin (K48-Ub) with or without Myc-DUSP22, followed by treatment with or without 25 μM MG132 for 4 h. For the identification of the UBR2-interacting proteins, immunocomplexes of Flag-tagged UBR2 were immunoprecipitated with anti-Flag antibody from lysates of HEK293T cells transfected with vector or Flag-UBR2 plasmid. For identification of the UBR2 dephosphorylation sites by DUSP22, Flag-tagged UBR2 were immunoprecipitated from Flag-UBR2-transfected HEK293T cells with or without Myc-DUSP22, followed by treatment with or without 25 μM MG132 for 4 h. For the identification of UBR2-induced Lck ubiquitination residues, immunocomplexes of Flag-tagged Lck were immunoprecipitated with anti-Flag antibody from lysates of Jurkat T cells co-transfected with Myc-UBR2 and Flag-Lck, followed by anti-CD3 antibody stimulation. Protein bands with higher molecular weights (>150 kDa) and lower molecular weights (<150 kDa) were collected from Instant Blue (GeneMark)-stained SDS-PAGE gels. Gel-extracted proteins were digested with trypsin and subjected to LC–MS/MS analyses by LTQ-Orbitrap Elite hybrid mass spectrometer (for identifying DUSP22-interacting proteins and UBR2-interacting protein) or LTQ-Orbitrap Fusion hybrid mass spectrometer (for mapping UBR2 phosphorylated residues, UBR2 ubiquitinated residues, and Lck ubiquitinated residues) using approaches described previously[43,44]. The peptide data were analyzed by MASCOT MS/MS Ions Search (Matrix Science) under the following conditions: peptide mass tolerance, 20 ppm; fragment MS/MS tolerance, 0.6 Da; allow up to one missed cleavage; peptide charge, $2^+$, $3^+$, and $4^+$.

## In situ proximity ligation assay (PLA)

The HEK293T cells coexpressing the molecules (Flag-tagged UBR2 plus Myc-tagged DUSP22; Flag-tagged UBR2 plus either Myc-tagged CUL1 or F-box proteins; Flag-tagged UBR2 plus Myc-tagged βTrCP and either GFP-tagged DUSP22 or GFP-tagged DUSP22 (C88S); Flag-tagged UBR2, Flag-tagged UBR2 (S1694D/Y1697D), or Flag-tagged UBR2 (S1694A/Y1697F) plus Myc-tagged βTrCP and GFP-tagged DUSP22; Flag-tagged UBR2 plus Myc-tagged DUSP22 and GFP-tagged DUSP22 (C88S)) were fixed with 4 % formaldehyde for 1 h. The fixed cells were permeabilized by 0.1% Triton X-100 at room temperature for 2 h, and then washed twice by PBS, followed by blocking with PLA blocking buffer. The cells were hybridized with anti-Flag and anti-Myc antibodies, followed by rabbit- and mouse-specific secondary antibodies conjugated with oligonucleotides (PLA probes; Sigma-Aldrich). For human primary T cells, T cells were fixed with cold methanol for 2 min. The fixed cells were permeabilized by 0.1% triton X-100 at room temperature for 2 h, and then washed twice by PBS, followed by blocking with PLA blocking buffer. For murine primary T cells and human Jurkat T cells, the transcription factor buffer set (#562574, BD Pharmingen™) was used for fixation and permeabilization, followed by blocking with PLA blocking buffer. The K63-linked ubiquitination of the endogenous Lck proteins were recognized by anti-Lck and anti-Lys63-specific ubiquitin antibodies, followed by rabbit- and mouse-specific secondary antibodies conjugated with oligonucleotides (PLA probes; Sigma-Aldrich). The endogenous Lck Tyr394 phosphorylation was recognized by anti-Lck and anti-phospho-Lck (Y394) antibodies, followed by rabbit- and mouse-specific secondary antibodies conjugated with oligonucleotides (PLA probes; Sigma-Aldrich). After the reaction of ligation and amplification, the PLA signals from the pair of PLA probes in close proximity less than 40 nm were visualized[27]. For fluorescent signal in cells, images were detected by fluorescence microscope (DM2500; Leica Microsystems, Buffalo Grove, IL, USA), confocal microscope Leica TCS SP5, and Leica TCS SP5 II. Images were analyzed by LAS X software (v3.7.4). The intensity value was obtained from area multiplying by mean.

**a**

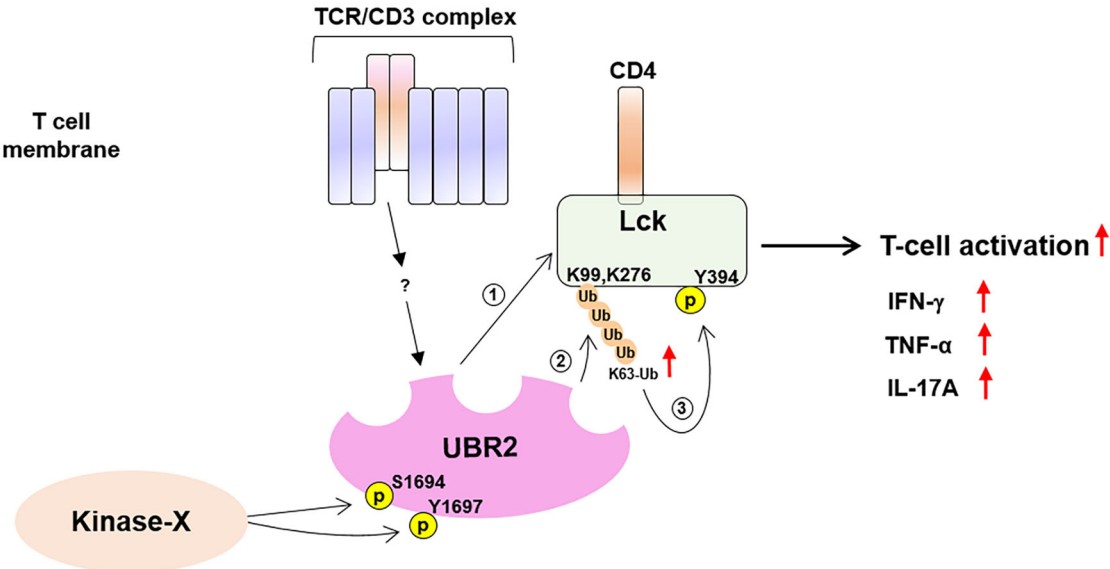

**TCR signaling turn-on (early activation) stage (Induction of inflammation)**

**b**

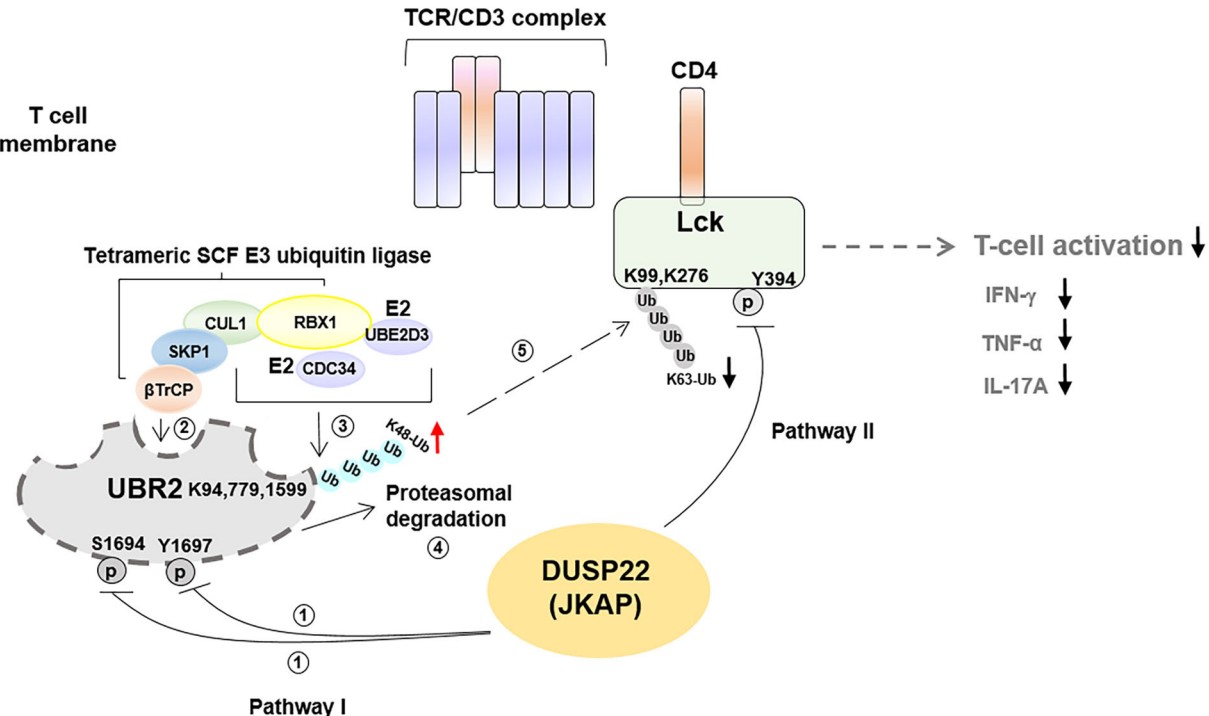

**TCR signaling turn-off (post activation) stage (Resolution of inflammation)**

## IHC

Kidneys, livers, and lungs were harvested from mice and fixed in 10% neutral-buffered formalin for 3–7 days at room temperature, and then transferred to 75% alcohol for further processing. Tissue processing, embedding, and H&E staining were performed by the NHRI Pathology Core.

## In vitro binding assay

Flag-UBR2 plasmid was transfected into HEK293T cells. Transfected cells were lysed in RIPA buffer (1% sodium deoxycholate, 1 mM EDTA, 150 mM NaCl, 1% Triton X-100, 0.1% SDS, 50 mM Tris-HCl) supplemented with 0.5% protease inhibitor cocktail, 1% phosphatase inhibitor cocktail, and 1 mM $Na_3VO_4$ at 4 °C for 30 min. Flag-tagged UBR2

**Fig. 8 | Schematic diagram of UBR2 regulation on Lck and DUSP22-induced dephosphorylation, K63-linked polyubiquitination, and degradation of UBR2.** **a** In TCR signaling initiation stage, UBR2 is phosphorylated at Ser1694 and Tyr1697 residues. Phosphorylated UBR2 interacts with Lck (step 1). UBR2 then facilitates Lck Lys63-linked polyubiquitination at Lys99 and Lys276 residues (step 2). TCR signaling-stimulated Lys63-linked polyubiquitination of Lck induces its Tyr394 phosphorylation (step 3), leading to enhanced T-cell activation, as well as increased proinflammatory cytokines. **b** In TCR signaling turn-off stage, DUSP22 inactivates Lck through both direct and indirect pathways. In pathway I, DUSP22 dephosphorylates UBR2 at Ser1694 and Tyr1697 residues (step 1). Dephosphorylation of

UBR2 results in the interaction of UBR2 with SKP1-CUL1-βTrCP (SCF E3 ubiquitin ligase complex) (step 2). Then, SCF E3 ubiquitin ligase complex induces Lys48-linked tri-ubiquitination at Lys94, Lys779, and Lys1599 residues of UBR2 (step 3), leading to UBR2 proteasomal degradation (step 4). Consequently, suppression of Lys63-linked Lck ubiquitination leads to Lck inactivation (step 5). Concurrently, in pathway II, DUSP22 directly dephosphorylates the activated Lck at Tyr394 residue, leading to Lck inactivation. The induction of UBR2 degradation and the induction of Lck inactivation by DUSP22-mediated dephosphorylation of UBR2 and Lck in pathway I and pathway II, respectively, attenuate TCR signaling and T-cell proinflammatory cytokines IFN-γ, TNF-α, and IL-17A levels.

proteins were immunoprecipitated with anti-Flag agarose beads in 1 ml of lysis buffer at 4 °C for 2 h. Flag-tagged UBR2 proteins were eluted by 3X Flag peptide (F4799, Sigma-Aldrich). GST-DUSP22 plasmid was transferred into *Escherichia coli* strain BL21 by transformation. GST-tagged DUSP22 proteins were precipitated by glutathione-conjugated sepharose beads (#17-0756-01, GE Healthcare), and then purified with glutathione (#G4251, Sigma) in GST elution buffer. Purified Flag-tagged UBR2 proteins were incubated with the GST-tagged DUSP22 proteins. The reaction mixtures were immunoprecipitated with anti-Flag agarose beads or anti-GST glutathione-sepharose at 4 °C for 2 h. The reaction mixtures were then subjected to immunoblotting analyses.

### In vitro ubiquitination assay
Purified Flag-tagged UBR2 proteins, SKP1-CUL1-βTrCP-RBX1 complex, E1 (0.1 μM UBE1), E2 (0.35 μM UBE2D3 or 1 μM CDC34), and His-ubiquitin (20 μM) were incubated at 37 °C for 1 h in 35 μl buffer containing 50 mM Tris (pH 7.5), 5 mM MgCl₂, 2 mM ATP, and 2 mM DTT. The reactant products were subjected to immunoblotting analyses. UBE1, UBE2D3, CDC34, and His-ubiquitin proteins were purchased from R&D Systems.

### In vitro ubiquitin E3 ligase assay in combination with in vitro kinase assay
Purified Myc-tagged UBR2 proteins, Flag-tagged Lck proteins, E1 (0.1 μM UBE1), E2 (0.98 μM UBE2N), and His-ubiquitin (20 μM) were incubated at 37 °C for 1 h in buffer containing 50 mM Tris (pH 8), 10 mM MgCl₂, 1 mM ATP, 1 mM DTT, and 50 mM NaCl. After in vitro E3 ligase assay, the reactants were incubated in kinase buffer containing 25 mM Tris-HCl (pH 7.5), 20 mM MgCl₂, and 5 mM beta-glycerophosphate at 37 °C for 0.5 h. The reaction mixtures were subjected to immunoblotting analyses. UBE1, UBE2N, and His-ubiquitin proteins were purchased from R&D Systems.

### In vitro phosphatase assay
Flag-UBR2, Myc-DUSP22, and Myc-DUSP22 (C88S) plasmids were individually transfected into HEK293T cells. Transfected cells were lysed in RIPA buffer. Flag-tagged UBR2 proteins were immunoprecipitated with anti-Flag agarose beads, and Myc-tagged DUSP22 proteins were immunoprecipitated with anti-Myc agarose beads in 1 ml of lysis buffer at 4 °C for 2 h. Flag-tagged UBR2 immunocomplexes were then incubated with Myc-tagged DUSP22 wild-type or mutant (C88S) proteins at 37 °C for 30 min in buffer containing imidazole (50 mM, pH 7.5) and DTT (10 mM). The reactant products were subjected to immunoblotting analysis.

### In vitro kinase assay
For kinase preparation, Myc-UBR2 plasmid plus either Flag-Lck, Flag-Lck (K99/276R), or Flag-Lck (Y394F) plasmid were co-transfected into Jurkat T cells. Transfected cells were lysed in RIPA buffer (1% sodium deoxycholate, 1 mM EDTA, 150 mM NaCl, 1% Triton X-100, 0.1% SDS, 50 mM Tris-HCl) supplemented with 0.5% protease inhibitor cocktail (#539134, EMD Millipore), 1% phosphatase inhibitor cocktail (#524629, EMD Millipore), and 1 mM Na₃VO₄ at 4 °C for 30 min. For

coimmunoprecipitation, 1 mg protein lysates were incubated with anti-Flag agarose beads (clone M2, Sigma-Aldrich) in 1 ml of lysis buffer at 4 °C for 1-2 h. The immunocomplexes were washed three times with lysis buffer. Flag-tagged wild-type, K99/276R, or Y394F Lck immunocomplexes from the lysates of UBR2-overexpressing Jurkat T cells were subjected to peptide elution by Flag peptides. For substrate preparation, Flag-Lck plasmid was transfected into HEK293T cells. Transfected cell lysates were extracted by RIPA buffer described above. Additional Flag-tagged wild-type Lck immunocomplex was isolated from the lysates of HEK293T cells using anti-Flag agarose beads. Purified Flag-tagged Lck proteins were incubated with the substrate Flag-tagged Lck for 40 min at 37 °C with 1 μM ATP in 60 μl kinase buffer. The reactant products were subjected to immunoblotting analyses.

### Purification of T cells
Peripheral blood was collected from mice or human individuals, and then red blood cells were lysed by ACK (ammonium-chloride-potassium) lysis buffer. To purify murine T cells, T cells were isolated by negative selection using a cocktail of biotin-conjugated antibodies against B220, CD49b, CD11b, CD11c, and TER-119 on a magnetic cell separation column (Miltenyi Biotec). To purify human T cells[45–47], peripheral blood T cells were negatively selected from 10 ml of whole blood from participants using a cocktail of biotin-conjugated antibodies against CD14, CD11b, CD11c, CD19, and CD235a on a magnetic cell separation column (Miltenyi Biotec).

### Enzyme-linked immunosorbent assays (ELISAs)
Serum levels of IFN-γ, and TNF-α were analyzed by individual ELISA kits purchased from eBioscience. The IL-17A levels were determined using an ELISA kit from BioLegend.

### Single-cell RNA sequencing analyses
Single-cell RNA sequencing was performed using peripheral blood T cells (total bulk T cells including CD4+ and CD8+ T cells) of 12- to 14-week-old UBR2 knockout and wild-type mice. The purified T cells were labeled by BD AbSeq Ab-Oligos reagents according to the manufacturer's protocol (BD Biosciences, #633793). Next, single-cell capture and cDNA synthesis were performed using the BD Rhapsody Single-Cell Analysis System (210966 v1.0). cDNA libraries were constructed by combining Sample Tag and BD AbSeq libraries (BD Biosciences, Cat 23-21752-00). cDNA libraries of individual cells were constructed using BD Rhapsody WTA Amplification Kit (Cat 633801). Paired-end sequencing was performed on a HiSeq X Ten sequencer (Illumina). Data analysis and quality control were performed following the BD Biosciences Rhapsody pipeline. Uniform Manifold Approximation and Projection (UMAP) generation were conducted with the R package Seurat (v3.0).

### T-cell development
Thymocytes, splenocytes, lymph node cells, and bone-marrow-derived cells were harvested from 5-week-old mice. For surface staining, cells were washed by PBS and were stained with antibodies at room temperature for 1 h (anti-CD3, anti-CD4, anti-CD8, anti-B220, and anti-IgM

antibodies). After incubation, cells were washed twice with PBS buffer to remove unbound antibodies. For intracellular Foxp3 staining, cells were permeabilized using Foxp3 Fix/Perm solution (BioLegend) and then incubated with diluted anti-Foxp3 antibodies in Foxp3 Perm buffer (BioLegend) at 4 °C for overnight. After incubation, cells were washed twice with Foxp3 Perm wash buffer (BioLegend) to remove unbound antibodies. For flow cytometry, data were acquired with a FACS CantoII (BD Biosciences) and analyzed with FlowJo (v10.8.1) analytical software.

## Crystal structure of human Lck protein
The Lck monomer structure model (63 to 509 a.a.) was generated by online homology modeling server, SWISS-MODEL[48,49]. The structure template was referred to hematopoietic cell kinase (Hck, PDB ID 1QCF, https://www.rcsb.org/structure/1QCF)[50] from RCSB Protein Data Bank (RSCB PDB, https://www.rcsb.org/). Lck protein sequence (UniProt ID P06239) was aligned to chicken Src (UniProt ID P00523) by using the online server Clustal Omega Multiple Sequence Alignment[51]. The interaction sites between SH3 domain, linker and catalytic domain according to the chicken Src structure (PDB ID 2PTK, https://www.rcsb.org/structure/2PTK)[36,37] were mapped on the aligned Lck sequence. The ubiquitin structure (PDB ID 3HMH, https://www.rcsb.org/structure/3HMH) was downloaded from RSCB PDB[52]. All the structures were operated by UCSF Chimera molecular structure interactive visualization and analysis program[53], followed by Visual Molecular Dynamics (VMD) molecular visualization program (http://www.ks.uiuc.edu/).

## T-cell activation assays
Human Jurkat T cells were stimulated with 5 μg/ml of anti-CD3 (clone OKT3) for the appropriate time at 37 °C. For supernatant collection, purified murine T cells were costimulated with 20 μg/ml of plate-bound anti-CD3 (clone 145-2C11)/anti-CD28 antibodies (clone 37.51). For PLA assay, purified T cells were stimulated with 3 μg/ml of biotin-conjugated anti-CD3 (clone 500A2, Cat 553239, Lot 0314175) plus 3 μg/ml of streptavidin (Sigma, Cat 85878-5MG, Lot 049k8616v).

## T-cell differentiation assays
In vitro differentiation assays of Th1, Th17, and Treg cells were performed using the methods described previously[47].

## EAE induction and MOG restimulation
Mice used in each experiment were 4- to 12-month-old female littermates. Experimental autoimmune encephalomyelitis (EAE) was induced by subcutaneous injection of mice (four per group) with 200 μg myelin oligodendrocyte glycoprotein (MOG) peptides (amino acids 35–55: MEVGWYRSPFSRVVHLYRNGK; Genemed Synthesis)[54] emulsified in CFA (Chondrex). Mice were also intraperitoneally injected with 200 ng pertussis toxin (List Biological Laboratories) on days 0, 1, and 2. Antigen-specific T cells were isolated from lymph nodes of MOG-immunized mice at day 14. Antigen-specific T cells were cultured in the presence of 0, 20, 50 μg/ml MOG for 72 h. Culture supernatants from MOG-restimulated T cells were analyzed by ELISA.

## Statistical analysis and reproducibility
Statistical analyses were performed by using Excel (v16.7) or BD SEQ-GEQ (v1.8.0). The statistical significance between two unpaired groups was analyzed using one- or two-tailed Student's t test. $P$ values of less than 0.05 were considered statistically significant. Symbols of $p$ values by one-tailed Student's t test represent $^{§}p < 0.05$ (Fig. 7d, middle). Symbols of $p$ values by two-tailed Student's t test represent $^{*}p < 0.05$, $^{**}p < 0.01$, $^{***}p < 0.001$, and $^{****}p < 0.0001$ (Figs. 5g, 5h, 7b, 7c, and 7i). Wilcoxon's rank-sum test was used to analyze violin plots. Symbols of $p$ values by Wilcoxon's rank-sum test represent $^{*}P < 0.05$, $^{**}P < 0.01$, $^{***}P < 0.001$, and $^{****}P < 0.0001$ (Fig. 5g). For reproducibility,

experiments were performed 3 independent times with $n = 6$ (IFN-γ and IL-17A) or $n = 7$ (TNF-α) (Fig. 5h); 6 independent times with $n = 10$ (Fig. 7b-c); 3 independent times with $n = 6$ (wild-type), $n = 4$ (DUSP22 KO), or $n = 2$ (DUSP22/UBR2 double KO) (Fig. 7d); 3 independent times with $n = 6$ (Fig. 7i). The data shown are representatives of 2 independent experiments (Fig. 2c and e; Fig. 6d, i, and j; Fig. 7e) and 3 independent experiments (Fig. 1b–d, f, g–m; Fig. 2a, b, f, and g; Fig. 3b–i; Fig. 4a, c–f; Fig. 5b, c; Fig. 6a–c, f–h; Fig. 7a, f–h). Data are presented as mean values ± SEM (Figs. 5h, 7b, 7c, 7d, 7i).

## Reporting summary
Further information on research design is available in the Nature Portfolio Reporting Summary linked to this article.

## Data availability
All data are available in the main text or the supplementary materials. The raw single cell RNA sequencing data generated in this study have been deposited in the Sequence Read Archive database under accession code of SRR25120110. The raw proteomic mass data generated in this study have been deposited in the ProteomeXchange Consortium via the PRIDE proteomics IDEntifications database under accession codes of PXD043490, PXD043463, PXD043457, and PXD043454. The protein structure was modeled by SWISS-MODEL web server and visualized by Visual Molecular Dynamics (VMD) molecular visualization program (http://www.ks.uiuc.edu/). The chicken Src structure (PDB ID 2PTK) was download from [https://www.rcsb.org/structure/2PTK]. The ubiquitin structure (PDB ID 3HMH) was downloaded from RSCB PDB [https://www.rcsb.org/structure/3HMH]. Source data are provided with this paper.

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

## Acknowledgements

The authors thank the Transgenic Mouse Core (NHRI, Taiwan) for generation of DUSP22 knockout mice and UBR2 knockout mice. The authors thank the NHRI Laboratory Animal Center for mouse housing. The authors thank the Institute of Biological Chemistry of Academia Sinica for mass spectrometry analyses. The authors thank the core facilities of NHRI for confocal microscopy and fluorescence microscope. The

authors thank members of Tan Lab for technical assistance, including Ms. Jhih-Yu Yang and Chia-Hsin Hsueh for in vitro binding assays, Mr. Pu-Ming Hsu for immunoprecipitation assays, Yu-Zhi Xiao for human patients PLA assays, and Ms. Ching-Yi Tsai for ELISAs. The authors also thank Dr. Hsien-Yi Chiu and Dr. Ming-Han Chen for providing clinical samples. This work was supported by grants from NHRI (IM-112-SP-01 to T.-H.T.) and the National Science and Technology Council, Taiwan (MOST-107-2314-B-400-008, and MOST-110-2320-B-400-018, NSTC-111-2320-B-400-003, and NSTC-112-2320-B-400-023) to T.-H.T.; MOST-108-2628-B-400-001 and NSTC-112-2320-B-400-024 to H.-C.C.). T.-H.T. is a Taiwan Bio-Development Foundation (TBF) Endowed Chair Professor in Biotechnology.

## Author contributions

H.-C.C., Y.-C.S, and H.-F.C. designed and performed experiments, analyzed and interpreted data, and wrote the manuscript; C.-Y.W. and Y.-R.C. performed experiments; C.-W.W. analyzed the crystal structure of human Lck protein; T.-H.T. conceived of the study, supervised experiments, and wrote the manuscript.

## Competing interests

The authors declare no competing interests.
