## [Peer Review File · Nature Communications]

The phosphatase DUSP22 inhibits UBR2-mediated K63-ubiquitination and activation of Lck downstream of TCR signallingREVIEWER COMMENTS

Reviewer #1 (Remarks to the Author):

The authors have elucidated a novel molecular mechanism by which UBR2 mediates T-cell activation via the K63 ubiquitination and activation of Lck. In more detail, during the TCR signaling turn-on stage, UBR2 is stabilized, as judged by its phosphorylation at S1694 and Y1697, which can induce the K63-linked ubiquitination and auto-phosphorylation of Lck, resulting in T-cell activation. In contrast, during the TCR signaling turn-off stage, JKAP/DUSP22 dephosphorylates UBR2 S1694 and Y1697, leading to SCF-CUL1- β TrCP-mediated ubiquitination and subsequent proteasomal degradation of UBR2. As a result, Lck is inactivated, and T-cell activation is inhibited. Finally, the authors demonstrated the biological relevance of their findings in systemic lupus erythematosus (SLE). The study has been conducted logically and thoroughly, and the findings are novel and significant. Therefore, I recommend this manuscript for publication in Nature Communications. However, there are minor issues that need to be addressed in order to perfect the manuscript.

Detailed comments:

Fig. 2f: The control for Flag-UBR2 (K94/779/1599R) alone is missing.

Fig. 3a: Were β TrCP tryptic peptides, in addition to CUL1 peptides, identified by LC-MS/MS? If not, why not?

Fig. 3d: In Fig. 3d figure legend, "Flag-CUL1" should be corrected to "Myc-JKAP."

Fig. 3g: In Fig. 3g figure legend, "left and right panel" should be corrected to "top and bottom panel."

Fig. 4b: Please add blots showing inputs.

Fig. 6a: In order for Lck to be activated, JKAP's interaction with Lck should decrease during α -CD3 stimulation. However, the results showed that JKAP-Lck interaction is increasing. Additionally, the level of endogenous UBR2 does not respond to α -CD3 stimulation, which is not consistent with their working model.

Fig. 6a & Suppl. fig. 7c: It is not clear if LCK-UBR2 interaction precedes Lck and JKAP interaction.

Fig. 6b: it would be nice to contrast the result by adding K48-linked ubiquitination as a negative control.

Fig. 6i: Please add blots showing inputs.

Fig. 6f: It will be better if there was a K99/276R condition in addition to only K99R or K276R.

Fig. 7a: Please show UBR and JKAP western blots.

Fig. 7h: Please examine if UBR2 is upregulated in SLE by performing western blotting.

Fig. 8: I wonder why UBR2 is needed if JKAP can directly dephosphorylate Lck and suppress its activation (pathway II). There is a lack of evidence from this study to support pathway II.

Reviewer #2 (Remarks to the Author):

This study provides insights into the molecular mechanism by which JKAP (DUSP22) regulates UBR2 protein levels and how UBR2 influences T cell function and Lck activation. It sheds light on the complex interplay between these proteins and their significance in immune cell regulation. The authors provide a great amount of data that is the result of extensive work. I believe that the work can be of significance to the field of T cell biology and highlights the importance of the DUSP family of phosphatases.

However, still several points should be addressed.

Major comments:

1. The introduction feels like a list of things that each molecule does. I believe it should be more connected, explaining better why the authors were focused on this problem.

2. Most of the studies were performed in HEK293T cells, which are not the most appropriate model to study T cell signalling and function. At least the key experiments in Figures 1-4 should be replicated using T cell lines such as Jurkat or primary T cells when possible.

3. The interaction between mutant JKAP with UBR2 seems to be higher than that of normal JKAP (Fig 1j and k). Any explanation?
4. Figure 1L data (PLA) is not very convincing. One would expect to see more specific and higher signal. Also, very few cells seem to be positive.
5. I believe Suppl. Fig 4c is important and should be in main figure 3
6. In Fig. 3d, JKAP overexpression alone does not induce UBR2 degradation. Why does JKAP suddenly need bTrCP overexpression to induce degradation of UBR2? This contradicts the key data shown in Fig 1c where overexpression of JKAP is enough to induce UBR2 degradation. Also, legend for Fig 3d is wrong since it mentions CUL1 instead of JKAP.
7. Relative to the previous point, in Fig 3b a control with only Flag-UBR2 is missing (without Myc-JKAP or Flag-CUL1)
8. Changes in p-Ser and p-Tyr phosphorylation levels in Figure 4b are very subtle to draw a conclusion.
9. Relative to the previous point, this paper heavily relies on western blot images. Maybe protein quantification would help some of the blots. Just as example, in Supplementary Figure 7, Lck, PLCgamma and ERK phosphorylation is reduced in UBR2-ko T cells, but total Lck, PLCgamma and ERK protein levels are also reduced, so it is difficult to be sure that this is a phosphorylation effect.
10. Figure 4c suggests that JKAP dephosphorylates UBR2 at three different sites, Ser1694, Ser476 and Tyr1697. However, in Figure 4d the UBR2 mutant S476D is still degraded. Does it mean that the phosphorylation at S476 is not important for JKAP activity?
11. Data from Figure 4d to 4g can be very confusing for the non-expert reader. If I understood correctly, explaining that the phosphomimetic mutant resembles the phosphorylated state and therefore will not be efficiently ubiquitinated and targeted for degradation by the bTrCP complex while the non-phosphomimetic mutant (S1694A/Y1697F) resembles the WT UBR2 after dephosphorylation by JKAP, would help to clarify the results.
12. Figure 5: the reviewer acknowledges the author's efforts in creating UBR2 knockout mice and performing scRNAseq. However, the T cell analyses fall a little short. First, this is a global knockout. To claim a specific effect of UBR2 on T cells, a conditional knockout would have been ideal. If not, at least stimulation of unexperienced naïve T cells should have been performed (Figure 5h includes total bulk T cells, both CD4 and CD8, am I correct?). Also, some kind of in vivo experiment showing that UBR2-ko naïve T cells have less inflammatory potential would have been nice. Or Treg assays to show they are perhaps more susceptible to Treg suppression?. What about proliferation? Or differentiation into Th subsets? There are a lot of functional assays that could be done to improve this part of the paper.
13. Figure 6: I don't see why the authors chose to show data on Jurkat T cells as main figure and data on primary T cells as supplementary (Suppl. Fig 7). In my opinion, data on primary T cells (even if they are murine) is always more relevant. For example, I don't see a significant induction of K63-Ub upon anti-CD3 stimulation in Figure 6b, there is a better response in primary T cells (Suppl. Figure 7d).
14. PLA data on figure 6c is not convincing. Only 1 cell out of 7 shows positive signal on a high magnification image. Do the authors have positive and negative controls to show specificity of the antibodies? Also, lower magnification pictures could help to have a better idea.
15. Also PLA data on Fig. 7b, there is little signal even in JKAP KO.
16. In Fig 7, I would show the EAE phenotype of JKAP and JKAP/UBR2 mice. One would expect to see severe EAE in JKAP mice and attenuation in the double knockout. Also, the use of UBR2+/- mice in Fig 7j is confusing. Why not use also -/- like in the rest of the figure?

17. Where is the evidence for the proposed pathway II in Figure 8? Data in Figure 6 and 7 show that UBR2 is required for Y394 phosphorylation, so the increased phosphorylation of Lck in the absence of JKAP could be due to overactivation of UBR2. Maybe I missed it, but I don't see evidence of a direct dephosphorylation of Lck by JKAP.

Minor comments:

1. The phrase "the interaction of JKAP or JAKP mutant with UBR2 was detected by reciprocal coimmunoprecipitation assay" (page 7) is repetitive (same thing is said a few lines before).
2. In page 16, "Base on the individual marker genes of various T-cell subsets (Supplementary Fig. 5c)". Suppl Fig 5c shows FACS analysis of B cells.
3. Suppl. Figure 6 is referenced in the text after Suppl. Fig 7. I would change the numbering.

Reviewer #3 (Remarks to the Author):

In the manuscript, Y-C. Shih et al. revealed the molecular mechanisms by which the phosphatase JKAP1/DUSP22 blocks UBR2-mediated Lck ubiquitination and activation, leading to reduced T lymphocyte activation and functions. Indeed, the authors demonstrate that the E3 ubiquitin ligase UBR2 controls Lck activation via Lys63-linked ubiquitination of Lck in early stage of TCR signaling. This is attenuated by JKAP1 in late stage of TCR signaling through UBR2 dephosphorylation leading to CRL1- β TrCP -induced UBR2 degradation by the proteasome. These mechanistic insights into the regulation of T lymphocyte activation provide new targets for pharmaceutical intervention in T lymphocyte-mediated autoimmune disease.

This is a well-designed and executed piece of work that is reported in a clear and concise manner, properly documented and embedded in the existing literature. The mechanisms are nicely demonstrated by a series of elegantly performed *in vitro* and *in vivo* experiments. Conclusions are well founded and carefully formulated. In the attempt to translate these data in humans, the Authors obtained evidence that UBR2 induces Lys63-linked ubiquitination of Lck in T lymphocytes of SLE patients. This excellent study that highlights the importance of ubiquitination events in the control of T lymphocyte activation is of interest for readers of *Nature Communications*.

Data on experimental models are solid. However, few issues remain to be addressed:

1. The lists of proteins identified by MS after immunoprecipitation of Myc-JKAP or Flag-UBR2 should be provided as supplemental materials. CUL 1 was coimmunoprecipitated with UBR2 although no direct interaction is expected. What about the other subunits of the E3 complex? Was β TrCP coprecipitated with UBR2?
2. Because MG132 is not a selective proteasome inhibitor, the authors cannot conclude that JKAP induces UBR2 proteasomal degradation. A selective proteasome inhibitor would have been a better choice.
3. To conclude for a direct interaction of JKAP with UBR2, *in vitro* binding assays should be performed with purified proteins. Please provide protein staining for assessment of the purity of the recombinant proteins. How was the GST-JKAP produced?
4. There are many F-box proteins acting as substrate recognition module. Why focusing only on β TrCP, SKP2, and FBXW7? From the data presented, we cannot exclude the contribution of other F-box proteins in controlling UBR2 stability.
5. The authors proposed that CRL1- β TrCP triggers UBR2 degradation at the TCR signaling turn-off stage thereby contributing to inflammation resolution. This is based on *in vitro* data or on data obtained by overexpression of recombinant proteins in HEK293T cells. The degradation of UBR2 driven by CRL1- β TrCP at late stage of T-cell activation should be demonstrated using T lymphocytes.
6. The sub-heading of the result section (page 15) and the title of Figure 5 indicate that UBR2 is involved in cell migration. However, no data supports this conclusion. Please clarify.

7. In Figure 5c, violin plots show the expression levels of cytokine genes under different UBR2 expression levels. Could you please clarify how UBR2 expression levels were measured and provide these values? Why UBR2 transcripts did not appear to be differentially expressed in control vs UBR2 KO T lymphocytes.
8. The enhanced phosphorylation of Lck in JKAP knockout T lymphocytes after TCR engagement is not obvious. Quantification of independent experiments should be provided.
9. In the images of PLA provided in Figures 3, 4, 6 and 7, the sizes of the scale bars are barely readable and should be removed from the images. Why not using the same magnifications for all the images of a given figure panel?
10. In Figure 7j, statistical analyses are needed.

Responses to Reviewer #1:

Overall Comment. The authors have elucidated a novel molecular mechanism by which UBR2 mediates T-cell activation via the K63 ubiquitination and activation of Lck. In more detail, during the TCR signaling turn-on stage, UBR2 is stabilized, as judged by its phosphorylation at S1694 and Y1697, which can induce the K63-linked ubiquitination and auto-phosphorylation of Lck, resulting in T-cell activation. In contrast, during the TCR signaling turn-off stage, JKAP/DUSP22 dephosphorylates UBR2 S1694 and Y1697, leading to SCF-CUL1- β TrCP-mediated ubiquitination and subsequent proteasomal degradation of UBR2. As a result, Lck is inactivated, and T-cell activation is inhibited. Finally, the authors demonstrated the biological relevance of their findings in systemic lupus erythematosus (SLE). The study has been conducted logically and thoroughly, and the findings are novel and significant. Therefore, I recommend this manuscript for publication in Nature Communications. However, there are minor issues that need to be addressed in order to perfect the manuscript.

Response: We appreciate the encouraging and constructive comments from Reviewer #1.

Comment 1. Fig. 2f: The control for Flag-UBR2 (K94/779/1599R) alone is missing.

Response: Per Reviewer's comment, we have included the control for Flag-UBR2 (K94/779/1599R) alone (Fig. 2f). Moreover, we have performed the experiment using Jurkat T cells instead of HEK293T cells (Fig. 2f, page 9, lines 18-19).

Comment 2. Fig. 3a: Were β TrCP tryptic peptides, in addition to CUL1 peptides, identified by LC-MS/MS? If not, why not?

Response: We did not detect any β TrCP tryptic peptides using LC-MS/MS analysis; this may be due to the limitation of mass spectrometry sensitivity or the inadequate β TrCP peptide lengths generated by trypsin digestion (page 11, lines 7-9).

Comment 3. Fig. 3d: In Fig. 3d figure legend, "Flag-CUL1" should be corrected to "Myc-JKAP."

Fig. 3g: In Fig. 3g figure legend, "left and right panel" should be corrected to "top and bottom panel."

Response: Per Reviewer's comment, we have revised the figure legend of Fig. 3d (now Fig. 3e, page 51, line 20) and Fig. 3g (now Fig. 3i, page 52, line 17).

Comment 4. Fig. 4b: Please add blots showing inputs.

Response: Per Reviewer's comment, we have added the loading inputs to Fig. 4b (now Fig. 4a).

Comment 5. Fig. 6a: In order for Lck to be activated, JKAP's interaction with Lck should decrease during α -CD3 stimulation. However, the results showed that JKAP-Lck interaction is increasing. Additionally, the level of endogenous UBR2 does not respond to α -CD3 stimulation, which is not consistent with their working model.

Response: We would like to clarify that the interaction between Lck and JKAP is induced in the TCR signaling turn-off stage, resulting in JKAP-induced dephosphorylation and inactivation of Lck¹⁴ (page 19, lines 2-4). In contrast, the interaction between Lck and UBR2 is increased during TCR signaling turn-on stage, leading to UBR2-induced Lys63-linked ubiquitination and activation of Lck (page 19, lines 1-2). It is

likely that either the catalytic activity or the substrate (Lck)-binding ability of UBR2 could be induced during the TCR signaling turn-on stage (page 19, lines 4-6); however, it is beyond the scope of this study. Ref #14: Li, J.-P. et al. The phosphatase JKAP/DUSP22 inhibits T-cell receptor signalling and autoimmunity by inactivating Lck. *Nat. Commun.* 5, 1-13 (2014).

Comment 6. Fig. 6a & Suppl. fig. 7c: It is not clear if LCK-UBR2 interaction precedes Lck and JKAP interaction.

Response: Per Reviewer's comment, we have clarified that the interaction between the endogenous Lck and UBR2 proteins in murine primary T cells was higher at 0 min and 5 min during α -CD3 stimulation (during TCR signaling turn-on stage) compared to those at 10 min, and the Lck-UBR2 interaction in murine primary T cells was significantly decreased at 10 min (during TCR signaling turn-off stage) (Fig. 6a). In contrast, the Lck-JKAP interaction was significantly induced at 10 min (during TCR signaling turn-off stage) (Fig. 6a). These results suggest that the Lck-UBR2 interaction precedes the Lck-JKAP interaction during the course of α -CD3 signaling. Immunoprecipitation of lysates from Lck-overexpressing Jurkat T cells showed similar results (Supplementary Fig. 6c) (page 17, line 17 to page 18, line 2).

Comment 7. Fig. 6b: it would be nice to contrast the result by adding K48-linked ubiquitination as a negative control.

Response: Per Reviewer's comment, we have added UBR2 K48-linked ubiquitination (+MG132) during TCR signaling turn-on stage (Supplementary Fig. 6e) as a negative control for Fig. 6b (now Supplementary Fig. 6d).

Comment 8. Fig. 6i: Please add blots showing inputs.

Response: Per Reviewer's comment, we have added the inputs in Fig. 6i.

Comment 9. Fig. 6f: It will be better if there was a K99/276R condition in addition to only K99R or K276R.

Response: Per Reviewer's comment, we have added UBR2 (K99/276R) to Fig. 6f. Immunoprecipitation and Western blotting analysis showed that individual Lck (K99R) mutation, Lck (K276R) mutation, and Lck (K99/276R) double mutations significantly alleviated UBR2-induced Lys-63-linked ubiquitination of Lck in HEK293T cells (Fig. 6f, page 20, lines 1-3).

Comment 10. Fig. 7a: Please show UBR and JKAP western blots.

Response: Due to the extremely low breeding rate of dKO mice, the purified T cells of dKO mice were not sufficient for additional immunoblotting of UBR2 and JKAP proteins in Fig. 7a. Instead, we have shown UBR2 and JKAP Western blots in a separate figure (Supplementary Fig. 8d, page 21, lines 22-23).

Comment 11. Fig. 7h: Please examine if UBR2 is upregulated in SLE by performing western blotting.

Response: Per Reviewer's comment, we have performed Western blotting of UBR2, p-Lck, JKAP, and Lck proteins using T cells of SLE patients. We found that UBR2 and p-Lck protein levels were upregulated, whereas JKAP protein levels were decreased, in T cells of SLE patients (Fig. 7f, page 22, lines 19-22).

Comment 12. Fig. 8: I wonder why UBR2 is needed if JKAP can directly dephosphorylate Lck and suppress its activation (pathway II). There is a lack of evidence from this study to support pathway II.

Response: We would like to clarify that in TCR signaling turn-on (early activation) stage, UBR2 plays a stimulatory role in the activation of Lck via Lys63-linked ubiquitination of Lck, leading to enhanced T-cell activation (Fig. 8a, page 26, lines 8-10). In contrast, JKAP can inhibit Lck in TCR signaling turn-off stage indirectly through pathway I and directly through pathway II. In pathway I, JKAP dephosphorylates UBR2, resulting in the SKP1-CUL1- β TrCP complex-induced UBR2 degradation and subsequent inhibition of UBR2-mediated Lck activation (Fig. 8b). In pathway II, JKAP directly dephosphorylates the activated Lck, leading to the obliterate of Lck activation¹⁴ (Fig. 8b, page 26, lines 10-15). The pathway II has been demonstrated in our previous publication (*Nature Communications*. 5, 1-13, 2014)¹⁴.

Ref #14: Li, J.-P. et al. The phosphatase JKAP/DUSP22 inhibits T-cell receptor signalling and autoimmunity by inactivating Lck. *Nat. Commun.* 5, 1-13 (2014).

Responses to Reviewer #2:

Overall Comment. This study provides insights into the molecular mechanism by which JKAP (DUSP22) regulates UBR2 protein levels and how UBR2 influences T cell function and Lck activation. It sheds light on the complex interplay between these proteins and their significance in immune cell regulation. The authors provide a great amount of data that is the result of extensive work. I believe that the work can be of significance to the field of T cell biology and highlights the importance of the DUSP family of phosphatases. However, still several points should be addressed.

Response: We appreciate the encouraging and constructive comments from Reviewer #2.

Major comments:

Comment 1. The introduction feels like a list of things that each molecule does. I believe it should be more connected, explaining better why the authors were focused on this problem.

Response: Per Reviewer's comment, we have revised the Introduction section accordingly (page 3, lines 14-23 and page 4, lines 15-21).

Comment 2. Most of the studies were performed in HEK293T cells, which are not the most appropriate model to study T cell signalling and function. At least the key experiments in Figures 1-4 should be replicated using T cell lines such as Jurkat or primary T cells when possible.

Response: Per Reviewer's comment, we have performed the key experiments using Jurkat T cells (Fig. 11, Supplementary Fig. 1a, Fig. 2f, Fig. 3b, Fig. 3e, Fig. 3h, and Fig. 4d (right panel)).

Comment 3. The interaction between mutant JKAP with UBR2 seems to be higher than that of normal JKAP (Fig 1j and k). Any explanation?

Response: We have clarified that "As expected, the levels of coimmunoprecipitated UBR2 proteins by JKAP "substrate-trapping (C88S) mutant²⁶" was higher than that of wild-type JKAP" (page 6, lines 19-21).

Ref #26: Lountos, G.T., Cherry, S., Tropea, J.E. & Waugh, D.S. Structural analysis of human dual-specificity phosphatase 22 complexed with a phosphotyrosine-like substrate. *Acta Crystallogr. F Struct. Biol. Commun.* 71, 199-205 (2015).

Comment 4. Figure 1L data (PLA) is not very convincing. One would expect to see more specific and higher signal. Also, very few cells seem to be positive.

Response: Per Reviewer's comment, we have added new PLA data using Jurkat T cells (Fig. 11). The data showed strong PLA signals of JKAP-UBR2 interaction in MG132-treated Jurkat T cell but not in other negative control T cells (Fig. 11, page 7, lines 2-4).

Comment 5. I believe Suppl. Fig 4c is important and should be in main figure 3

Response: Per Reviewer's comment, we have moved supplementary Fig. 4c to main Fig. 3c.

Comment 6. In Fig. 3d, JKAP overexpression alone does not induce UBR2 degradation. Why does JKAP suddenly need bTrCP overexpression to induce degradation of UBR2? This contradicts the key data shown in Fig 1c where overexpression of JKAP is enough to induce UBR2 degradation. Also, legend for Fig 3d is wrong since it mentions CUL1 instead of JKAP.

Response: Per Reviewer's comment, we have clarified that "UBR2 protein levels were decreased by β TrCP overexpression in the presence of "suboptimal" amounts of JKAP" (page 11, lines 10-12). The typos in Fig. 3d (now Fig. 3e) have been corrected.

Comment 7. Relative to the previous point, in Fig 3b a control with only Flag-UBR2 is missing (without Myc-JKAP or Flag-CUL1)

Response: Per Reviewer's comment, we have added a control with only Flag-UBR2 to Fig. 3b. Moreover, we have performed this experiment using Jurkat T cells instead of HEK293T cells (Fig. 3b, page 10, lines 7-10).

Comment 8. Changes in p-Ser and p-Tyr phosphorylation levels in Figure 4b are very subtle to draw a conclusion.

Response: Per Reviewer's comment, we have provided the data of improved quality for Fig. 4b (now Fig. 4a).

Comment 9. Relative to the previous point, this paper heavily relies on western blot images. Maybe protein quantification would help some of the blots. Just as example, in Supplementary Figure 7, Lck, PLCgamma and ERK phosphorylation is reduced in UBR2-ko T cells, but total Lck, PLCgamma and ERK protein levels are also reduced, so it is difficult to be sure that this is a phosphorylation effect.

Response: Per Reviewer's comment, we have provided quantification of p-Lck, p-PLC γ , and p-ERK proteins (normalized to their total-form proteins) at the bottom of individual panels in the Supplementary Fig. 7a, b (now Supplementary Fig. 6a, b).

Comment 10. Figure 4c suggests that JKAP dephosphorylates UBR2 at three different sites, Ser1694, Ser476 and Tyr1697. However, in Figure 4d the UBR2 mutant S476D is still degraded. Does it mean that the phosphorylation at S476 is not important for JKAP activity?

Response: Yes, we have clarified that "The data suggest that JKAP-induced UBR2 dephosphorylation at

Ser1694 and Tyr1697 residues, but not Ser476 residue, is responsible for JKAP-induced UBR2 proteasomal degradation” (page 14, lines 5-7).

Comment 11. Data from Figure 4d to 4g can be very confusing for the non-expert reader. If I understood correctly, explaining that the phosphomimetic mutant resembles the phosphorylated state and therefore will not be efficiently ubiquitinated and targeted for degradation by the bTrCP complex while the non-phosphomimetic mutant (S1694A/Y1697F) resembles the WT UBR2 after dephosphorylation by JKAP, would help to clarify the results.

Response: Per Reviewer’s comment, we have clarified that “phosphomimetic mutant (S1694D/Y1697D) of UBR2, resembling its phosphorylated state, would not be efficiently ubiquitinated and targeted for degradation by the SKP1-CUL1- β TrCP complex. In contrast, the phospho-deficient mutant (S1694A/Y1697F) of UBR2 resembles unphosphorylated UBR2 proteins after dephosphorylation by JKAP, ensuing its degradation by the SKP1-CUL1- β TrCP complex” (page 14, line 19 to page 15, line 1).

Comment 12. Figure 5: the reviewer acknowledges the author’s efforts in creating UBR2 knockout mice and performing scRNAseq. However, the T cell analyses fall a little short. First, this is a global knockout. To claim a specific effect of UBR2 on T cells, a conditional knockout would have been ideal. If not, at least stimulation of unexperienced naïve T cells should have been performed (Figure 5h includes total bulk T cells, both CD4 and CD8, am I correct?). Also, some kind of *in vivo* experiment showing that UBR2-ko naïve T cells have less inflammatory potential would have been nice. Or Treg assays to show they are perhaps more susceptible to Treg suppression?. What about proliferation? Or differentiation into Th subsets? There are a lot of functional assays that could be done to improve this part of the paper.

Response: (a) We have not yet generated UBR2 floxed mouse line.

(b) Yes, total bulk T cells including CD4⁺ and CD8⁺ T cells of mice were subjected to single-cell RNA sequence (page 37, lines 21-22).

(c) In the past 4 years, we obtained only 40 UBR2 homozygous knockout mice, including 38 male and 2 female mice, from 468 offspring mice (page 15, lines 14-15). Due to the extreme difficulty in the breeding of UBR2 homozygous knockout mice (see page 15, lines 12-15), we have only performed *in vitro* T-cell differentiation analysis using UBR2 homozygous knockout mice. We have added the following text “To study whether T-cell differentiation is affected by UBR2 knockout, *in vitro* differentiation assays were performed using splenic T cells of UBR2 knockout mice. Th1 and Th17, but not Treg, differentiation was decreased in T cells of UBR2 knockout mice compared to wild-type mice (Supplementary Fig. 5g). Collectively, these results suggested that UBR2 plays an important role in Th1/Th17-mediated inflammation” to the Results section (Supplementary Fig. 5g, page 16, line 19 to page 17, line 2).

Comment 13. Figure 6: I don’t see why the authors chose to show data on Jurkat T cells as main figure and data on primary T cells as supplementary (Suppl. Fig 7). In my opinion, data on primary T cells (even if they are murine) is always more relevant. For example, I don’t see a significant induction of K63-Ub upon anti-CD3 stimulation in Figure 6b, there is a better response in primary T cells (Suppl. Figure 7d).

Response: Per Reviewer’s comment, we have moved Supplementary Fig. 7d to main Fig. 6a.

Comment 14. PLA data on figure 6c is not convincing. Only 1 cell out of 7 shows positive signal on a high magnification image. Do the authors have positive and negative controls to show specificity of the antibodies? Also, lower magnification pictures could help to have a better idea.

Response: Per Reviewer's comment, we have replaced Fig. 6c with high-quality pictures and added negative controls using Lck-deficient Jurkat (J.Cam1.6 clone) T cells to Supplementary Fig. 6h, i, page 18, lines 18-22). Also, the PLA signals in Fig. 6c were too weak to be visualized in lower magnification pictures; alternately, we have added the data of three additional fields to Supplementary Fig. 6h, i.

Comment 15. Also PLA data on Fig. 7b, there is little signal even in JKAP KO.

Response: Per Reviewer's comment, we have replaced Fig. 7b with a high-quality figure.

Comment 16. In Fig 7, I would show the EAE phenotype of JKAP and JKAP/UBR2 mice. One would expect to see severe EAE in JKAP mice and attenuation in the double knockout. Also, the use of UBR2+/- mice in Fig 7j is confusing. Why not use also -/- like in the rest of the figure?

Response: We have clarified that female mice are required for optimal induction of EAE³⁵. Due to the lack of female JKAP/UBR2 double knockout mice, we characterized the T cells isolated from lymph nodes of female JKAP/UBR2 heterozygous knockout mice (page 23, lines 10-13).

Comment 17. Where is the evidence for the proposed pathway II in Figure 8? Data in Figure 6 and 7 show that UBR2 is required for Y394 phosphorylation, so the increased phosphorylation of Lck in the absence of JKAP could be due to overactivation of UBR2. Maybe I missed it, but I don't see evidence of a direct dephosphorylation of Lck by JKAP.

Response: Per Reviewer's comment, we have clearly described the mechanisms of JKAP-inhibited Lck activation and have added the evidence/reference for pathway II (page 26, lines 10-15).

Ref #14: Li, J.-P. et al. The phosphatase JKAP/DUSP22 inhibits T-cell receptor signaling and autoimmunity by inactivating Lck. *Nat. Commun.* 5, 1-13 (2014).

Minor comments:

1. The phrase "the interaction of JKAP or JAKP mutant with UBR2 was detected by reciprocal coimmunoprecipitation assay" (page 7) is repetitive (same thing is said a few lines before).

Response: Per Reviewer's comment, we have revised the sentence accordingly (page 6, lines 12-22).

2. In page 16, "Base on the individual marker genes of various T-cell subsets (Supplementary Fig. 5c)". Suppl Fig 5c shows FACS analysis of B cells.

Response: Per Reviewer's comment, the typos in Suppl Fig. 5c have been revised (page 15, lines 19-20).

3. Suppl. Figure 6 is referenced in the text after Suppl. Fig 7. I would change the numbering.

Response: Per Reviewer's comment, we have changed the numbering of Supplementary Fig. 6 and 7.

Responses to Reviewer #3:

Overall Comment. In the manuscript, Y-C. Shih et al. revealed the molecular mechanisms by which the phosphatase JKAP1/DUSP22 blocks UBR2-mediated Lck ubiquitination and activation, leading to reduced T lymphocyte activation and functions. Indeed, the authors demonstrate that the E3 ubiquitin ligase UBR2 controls Lck activation via Lys63-linked ubiquitination of Lck in early stage of TCR signaling. This is attenuated by JKAP1 in late stage of TCR signaling through UBR2 dephosphorylation leading to CRL1- β TrCP -induced UBR2 degradation by the proteasome. These mechanistic insights into the regulation of T lymphocyte activation provide new targets for pharmaceutical intervention in T lymphocyte-mediated autoimmune disease.

This is a well-designed and executed piece of work that is reported in a clear and concise manner, properly documented and embedded in the existing literature. The mechanisms are nicely demonstrated by a series of elegantly performed *in vitro* and *in vivo* experiments. Conclusions are well founded and carefully formulated. In the attempt to translate these data in humans, the Authors obtained evidence that UBR2 induces Lys63-linked ubiquitination of Lck in T lymphocytes of SLE patients. This excellent study that highlights the importance of ubiquitination events in the control of T lymphocyte activation is of interest for readers of Nature Communications. Data on experimental models are solid. However, few issues remain to be addressed:

Response: We appreciate the encouraging and constructive comments from Reviewer #3.

Comment 1. The lists of proteins identified by MS after immuno purification of Myc-JKAP or Flag-UBR2 should be provided as supplemental materials. CUL1 was coimmunoprecipitated with UBR2 although no direct interaction is expected. What about the other subunits of the E3 complex? Was β TrCP coprecipitated with UBR2?

Response: (a) Per Reviewer's comment, we have provided the JKAP- and UBR2-interacting proteins in ProteomeXchange database, no. PXD043490, PXD043463, PXD043457, and PXD043454.
(b) We did not detect any β TrCP tryptic peptides using LC-MS/MS analysis; this may be due to either the limitation of mass spectrometry sensitivity or the inadequate β TrCP peptide lengths generated by trypsin digestion (page 11, lines 7-9).

Comment 2. Because MG132 is not a selective proteasome inhibitor, the authors cannot conclude that JKAP induces UBR2 proteasomal degradation. A selective proteasome inhibitor would have been a better choice.

Response: Per Reviewer's comment, we have performed the experiment using the selective proteasome inhibitor carfilzomib. The data showed that JKAP-induced UBR2 proteasomal degradation was reversed by treatment of carfilzomib (Supplementary Fig. 1a, page 6, lines 9-10).

Comment 3. To conclude for a direct interaction of JKAP with UBR2, *in vitro* binding assays should be performed with purified proteins. Please provide protein staining for assessment of the purity of the recombinant proteins. How was the GST-JKAP produced?

Response: Per Reviewer's comment, we have performed *in vitro* binding assays using purified proteins (Fig. 1m). We have added the following text "GST-JKAP plasmid was transferred into *Escherichia coli*

strain BL21 by transformation. GST-tagged JKAP proteins were precipitated by glutathione-conjugated sepharose beads (#17-0756-01, GE Healthcare), and then purified with glutathione (#G4251, Sigma) in GST elution buffer.” to Materials and Methods section (page 34, line 21 to page 35, line 5). We have provided the full protein staining with Coomassie Blue in the Supplementary Fig. 9.

Comment 4. There are many F-box proteins acting as substrate recognition module. Why focusing only on β TrCP, SKP2, and FBXW7? From the data presented, we cannot exclude the contribution of other F-box proteins in controlling UBR2 stability.

Response: β TrCP, SKP2, and FBXW7 are the best-characterized F-box proteins within the CUL1 SCF ubiquitin ligase complexes (page 10, line 23 to page 11, line 1). We agree with Reviewer’s comment and added the following text “Although we could not rule out the contribution of additional F-box proteins in controlling UBR2 stability, our data suggest that β TrCP is a key F-box protein for JKAP-induced UBR2 degradation.” to the Results section (page 11, lines 12-14).

Comment 5. The authors proposed that CRL1- β TrCP triggers UBR2 degradation at the TCR signaling turn-off stage thereby contributing to inflammation resolution. This is based on in vitro data or on data obtained by overexpression of recombinant proteins in HEK293T cells. The degradation of UBR2 driven by CRL1- β TrCP at late stage of T-cell activation should be demonstrated using T lymphocytes.

Response: Per Reviewer’s comment, we have performed β TrCP or CUL1 shRNA knockdown in Jurkat T cells. The data showed that UBR2 protein levels were decreased at 15 min in the TCR signaling turn-off stage (Fig. 3h). In contrast, UBR2 degradation was blocked by shRNA knockdown of β TrCP or CUL1 in the TCR turn-off stage (Fig. 3h). These results suggested that UBR2 stability is reduced by the β TrCP-CUL1-containing E3 complex during the late stage of T-cell activation. (Fig. 3h, page 12, lines 4-9).

Comment 6. The sub-heading of the result section (page 15) and the title of Figure 5 indicate that UBR2 is involved in cell migration. However, no data supports this conclusion. Please clarify.

Response: We apologize for the inclusion of irrelevant text, which have been removed in the revised manuscript (Fig. 5, page 15, line 7 and page 54, line 13).

Comment 7. In Figure 5c, violin plots show the expression levels of cytokine genes under different UBR2 expression levels. Could you please clarify how UBR2 expression levels were measured and provide these values? Why UBR2 transcripts did not appear to be differentially expressed in control vs UBR2 KO T lymphocytes.

Response: Per Reviewer’s comment, we have clarified that mRNA levels of UBR2 and cytokines were detected as unique molecular identifier (UMI) counts (page 55, line 4). We have clarified that “Next, we studied whether UBR2 levels are correlated with the levels of inflammatory cytokines using scRNA-seq data. We analyzed these correlation using scRNA-seq data derived only from wild-type mice, because UBR2 knockout T cells still expressed nonfunctional UBR2 mRNAs containing 68-bp internal deletion” (page 16, lines 8-11).

Comment 8. The enhanced phosphorylation of Lck in JKAP knockout T lymphocytes after TCR

engagement is not obvious. Quantification of independent experiments should be provided.

Response: Per Reviewer's comment, we have provided quantification of p-Lck proteins in Fig. 7a.

Comment 9. In the images of PLA provided in Figures 3, 4, 6 and 7, the sizes of the scale bars are barely readable and should be removed from the images. Why not using the same magnifications for all the images of a given figure panel?

Response: Per Reviewer's comment, we used higher magnifications in the images to clearly show the PLA signals that were too small to be visualized in lower magnification pictures; therefore, we cannot remove all the scale bars due to the different magnifications for different pictures. Alternatively, we have increased the size of the text and used thicker lines as scale bars to enhance their readability in Fig. 4e, 6c, 6h, 7b, 7c, 7g, 7h, Supplementary Fig. 6h, i. Moreover, we have described the magnification of Fig. 3g, Supplementary Fig. 1b, 4c in the figure legend.

Comment 10. In Figure 7j, statistical analyses are needed.

Response: Per Reviewer's comment, we have provided statistical analysis in Fig. 7j.

REVIEWERS' COMMENTS

Reviewer #1 (Remarks to the Author):

The authors have addressed my major concerns. However, I noticed a minor mistake: Fig. 2f appears to be mislabeled. It should read K94/779/1599R instead of K99/779/1599R. Once this minor error is corrected, I believe the manuscript will be worthy of publication in Nature Communications.

Reviewer #2 (Remarks to the Author):

The authors have addressed all my comments and I believe the manuscript is suitable for publication at Nature Communications.

Reviewer #3 (Remarks to the Author):

The manuscript has been adequately revised to address my comments and the point by point reply is serious and adequate.

Reply to Reviewer #1:

Overall Comment. The authors have addressed my major concerns. However, I noticed a minor mistake: Fig. 2f appears to be mislabeled. It should read K94/779/1599R instead of K99/779/1599R. Once this minor error is corrected, I believe the manuscript will be worthy of publication in Nature Communications.

Reply: We appreciate the encouraging and constructive comments from Reviewer #1. Per Reviewer's comment, we have corrected the typos in Fig. 2f.

Reply to Reviewer #2:

Overall Comment. The authors have addressed all my comments and I believe the manuscript is suitable for publication at Nature Communications.

Reply: We appreciate the encouraging comments from Reviewer #2.

Reply to Reviewer #3:

Overall Comment. The manuscript has been adequately revised to address my comments and the point by point reply is serious and adequate.

Reply: We appreciate the encouraging comments from Reviewer #3.